# Rapid tuning shifts in human auditory cortex enhance speech intelligibility

Christopher R. Holdgraf[1], Wendy de Heer[2], Brian Pasley[1], Jochem Rieger[1], Nathan Crone[3], Jack J. Lin[4], Robert T. Knight[1,2,3] & Frédéric E. Theunissen[1,2]

Experience shapes our perception of the world on a moment-to-moment basis. This robust perceptual effect of experience parallels a change in the neural representation of stimulus features, though the nature of this representation and its plasticity are not well-understood. Spectrotemporal receptive field (STRF) mapping describes the neural response to acoustic features, and has been used to study contextual effects on auditory receptive fields in animal models. We performed a STRF plasticity analysis on electrophysiological data from recordings obtained directly from the human auditory cortex. Here, we report rapid, automatic plasticity of the spectrotemporal response of recorded neural ensembles, driven by previous experience with acoustic and linguistic information, and with a neurophysiological effect in the sub-second range. This plasticity reflects increased sensitivity to spectrotemporal features, enhancing the extraction of more speech-like features from a degraded stimulus and providing the physiological basis for the observed 'perceptual enhancement' in understanding speech.

[1] Helen Wills Neuroscience Institute, University of California, Berkeley, California 94720, USA. [2] Department of Psychology, University of California, Berkeley, California 94720, USA. [3] Department of Neurology, The Johns Hopkins University School of Medicine, Baltimore, Maryland 21205, USA. [4] UC Irvine Comprehensive Epilepsy Program, Department of Neurology, University of California, Irvine, California 92868, USA. Correspondence and requests for materials should be addressed to C.R.H. (email: choldgraf@berkeley.edu).

Auditory perception encompasses a sequence of feature extraction steps, with increasingly complex acoustic features extracted at each stage of neural processing[1,2]. Auditory neuroscientists have used synthetic and natural sounds as stimuli while recording the neural activity of single auditory neurons to investigate the nature of these computations. This research has led to an understanding of cortical auditory processing as a modulation filter bank[3]. At the level of auditory cortex, sounds are decomposed not only in frequency channels (as in the auditory periphery) but also in terms of joint spectral and temporal modulations. The filters in this modulation filter bank are the neurons' spectrotemporal receptive fields (STRFs)[4–6]. The decomposition of sounds into a modulation filter bank facilitates many tasks, including the discrimination of speech from non-speech[7] and the extraction of communication signals from noise[8].

Several studies have examined a STRF-based feature representation at different levels of the auditory hierarchy[5,9,10], but it is less understood if and how these representations interact with each other. For example, the presence of a higher-level response (such as the recognition of task-relevant stimuli) may alter the way that stimulus features are represented at lower levels in the auditory processing stream[11]. It has been shown that the tuning of auditory neurons changes during behavioural tasks[12–15], revealing that the STRFs describing this tuning are plastic. Further, neuroanatomical[16–18] and neurophysiological[19,20] research have highlighted the importance of top-down mechanisms for inducing such task-dependent STRF plasticity. These results were all obtained from single-unit recordings in animal models, and top-down manipulation was generally modulated with active attentional manipulations or task-relevant demands.

Human speech perception is another area in which top-down and bottom-up mechanisms are in constant interplay[21–23]. The act of understanding speech requires that auditory information entering the auditory periphery is interpreted through the lens of previous experience with natural sounds and language. It is assumed that this experience plays a role in shaping the cortical response to speech. Recent research using human electrophysiology has shown that experience with sound or contextual information about its content is correlated with differing patterns of low-frequency activity in both auditory and premotor cortex. For example, activity in the theta band of neural signals is reported to track the temporal structure in the speech envelope[24–26] and this tracking increases as noise levels are decreased in the speech stimulus[27]. In addition, power in theta and beta frequency bands have been implicated in top-down processing during speech perception[25]. It has been suggested that these signals reflect the brain's attempt to find relevant information in the speech signal, and to filter out noise or competing auditory streams[28]. While these approaches delineate differing patterns of neural activity that reflect top-down processes, they do not quantify changes in the spectrotemporal tuning of cortical activity, a feature representation that is believed to be encoded in auditory cortical neurons.

To investigate how contextual effects modulate auditory cortical activity, it is necessary to investigate the feature representations that are encoded in auditory brain areas. STRF models have been used as a standard for characterizing the tuning of neurons in primary auditory cortex[2]. Recent research has shown that STRF modelling may be applied to human electrocorticography (ECoG) to characterize the spectrotemporal tuning of cortical sites in response to speech[29–31] and to investigate plasticity in the auditory cortical response[32]. In particular, the high-frequency broadband (HFB; 70–150 Hz) component recorded with ECoG has both the spatial resolution to localize activity to discrete regions of the brain, and the temporal resolution to resolve excitation by the fine grained patterns of acoustic features.[33,34]. This permits using HFB activity to study the representation of spectrotemporal speech features in human auditory cortex and investigate how this representation changes during language processing.

Here, we perform a passive listening speech task in ECoG subjects. In this task, subjects hear degraded speech before and after experience with an unfiltered speech context. We first document that perceptual enhancement to the degraded sound is boosted after experience with the unfiltered speech, enabling speech comprehension. We then use STRF modelling techniques to investigate if this perceptual enhancement coincides with a shift in auditory cortical tuning to spectrotemporal speech features. We use regularized regression techniques to estimate a STRF for each recording electrode. It is estimated that the HFB activity of a single electrode reflects the activity of hundreds of thousands of neurons[35,36]. Thus, we effectively calculate an ensemble spectrotemporal receptive field, which we refer to as an eSTRF. We subsequently use this acronym to explicitly distinguish our results from those obtained with single auditory units. We show that providing an unfiltered speech context before a degraded speech stimulus causes an automatic, rapid shift in auditory cortical eSTRFs, enhancing their sensitivity to speech features. These findings provide evidence of an automatic mechanism in which experience with a contextually appropriate speech sentence causes behavioural perceptual enhancement for subsequent degraded speech signals, along with a tuning shift towards speech-specific spectrotemporal auditory features in auditory cortical areas.

## Results

**ECoG behavioural task**. A passive listening filtered speech task was used to study the neural response to degraded speech before and after hearing an unfiltered speech context. Filtered speech stimuli were created by filtering out portions of the modulation power spectrum (MPS) of each sentence (see Fig. 1, Methods; and Supplementary Audio 1) with low-pass filters. The corner frequency of each filter was chosen to render speech unintelligible by removing key spectral or temporal modulations[37]. ECoG subjects ($n = 7$) heard a filtered version of a speech utterance (hereafter described as the BEFORE condition), followed by an unfiltered version of the sound (MIDDLE condition), and finally by a repetition of the filtered version (AFTER condition). The first filtered speech presentation is incomprehensible, while the second filtered speech presentation is understandable due to experience with the unfiltered speech context. See Fig. 2 for a description of task design.

**Behavioural control study**. Due to limitations of the ECoG recording environment, it was not possible to obtain behavioural response data from ECoG patients, and a separate task was performed on control subjects to validate and quantify the perceptual effects generated in our stimuli sequences. In one task, subjects heard a single filtered version of each stimulus (with no unfiltered speech context), and were asked to type any words that they understood. The per cent correct was calculated for each sentence. Without any unfiltered speech context, subjects recognized $3.5 \pm 0.4\%$ of filtered speech words, replicating previous studies with the same filtering technique[37]. In a second task, subjects listened to filtered speech sentences along with a number of different context sentences, mimicking the 'filtered-unfiltered-filtered' structure of the behavioural task that the patients performed. Subjects typed out any words that they understood after the second presentation of the filtered speech sentence.

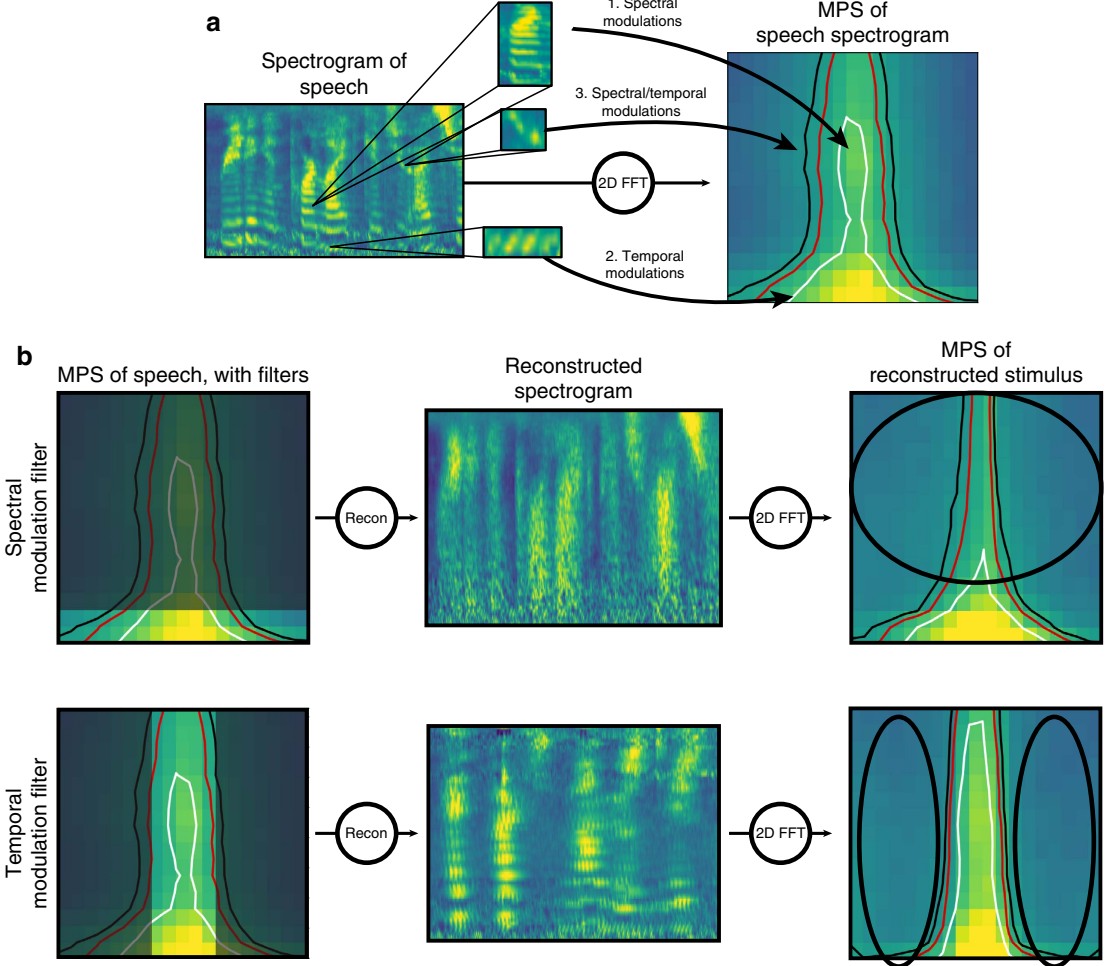

**Figure 1 | Stimulus creation by filtering MPS.** (**a**) The MPS describes the oscillatory patterns present in a time-frequency representation of sound. Left, the spectrogram of unfiltered speech is shown. Right, the MPS (calculated from a 2D FFT) is shown. Patterns in the spectrogram are reflected as power in temporal or spectral axes of the MPS. Rapid spectral fluctuations (for example, harmonic stacks from pitch, (1)) are represented near the middle/top of the MPS. Rapid temporal fluctuations (for example, plosives, (2)) are represented near the bottom/sides of the MPS. Joint spectral/temporal fluctuations (for example, rising pitch and phoneme changes, (3)) are represented in the upper corners of the MPS. (**b**) Left column: Filtered speech was created by filtering either spectral (top) or temporal (bottom) regions of the MPS space. MIDDLE column: spectrograms of the resulting filtered speech is shown. Right column: re-calculating the MPS on the filtered speech spectrogram shows that the MPS is now lacking power in the filtered regions.

Without any context, subjects understood 4.53 ± 0.82% words. When they were given a contextual sentence that was different from the filtered speech sentence, subjects understood 10.5 ± 1.3% words, representing the perceptual enhancement due to stimulus repetition or general activation of auditory streams involved in the processing of intact speech. When the contextual sentence was the same sentence as the filtered speech, subjects understood 77.7 ± 1.5% of words. As such, there was a roughly 67.2% increase in comprehension relative to hearing a different contextual sentence, representing the perceptual enhancement we focus on in this paper (two-sample $t$-test, $P = 1e-5$, df = 16). This perceptual enhancement reflects multiple speech processes, including the recent activation of the auditory stream in response to clean speech (as in the different sentence case) as well as the activation of cognitive areas involved in language processing resulting from speech comprehension (Fig. 2; Supplementary Fig. 1).

**HFB activity**. All analyses in this study were based on the HFB activity (70–150 Hz) of electrocorticographic (ECoG) recordings. This HFB signal is characterized by an increase in power across a large range of frequencies, and reflects local neuronal firing

within ~2 mm of each electrode, representing the combined activity of ~500,000 cortical neurons[33,36]. HFB can provide low-noise single trial evoked responses (see Fig. 6, as well as Supplementary Movies 1 and 2) that has been used for speech decoding and encoding models in humans[31,32], making it a good candidate for STRF modelling (Supplementary Fig. 4).

To define speech-selective electrodes, the mean post-stimulus HFB activity was first calculated for every speech trial. For each electrode, we used standard bootstrapping methods to calculate a bootstrap distribution of its mean evoked HFB activity across trials. The 0.5th percentile of this distribution was then calculated as a lower bound on post-stimulus activity (corresponding to a 99% confidence interval). This process was repeated for each electrode, and electrodes with a lower bound greater than 0 were defined as speech-selective. A subset of electrodes in each subject had significant responses to speech stimuli over baseline (confidence interval test across trials, see Fig. 3), generally centred around perisylvian regions. These electrodes are subsequently called speech-responsive (Speech-R) and made up 92 of 468 total electrodes (19.6%, see Supplementary Fig. 2). It should be noted that these electrodes did not show significant changes in activity over the course of the entire session (Supplementary

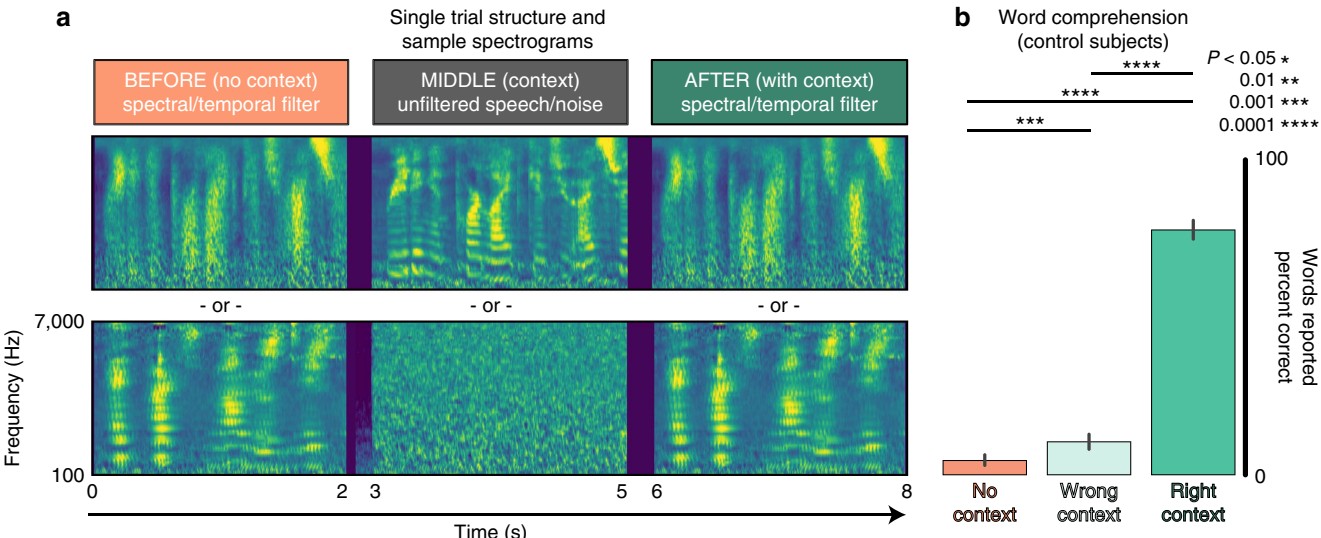

**Figure 2 | Behavioural task and speech intelligibility results.** (**a**) Trials consisted of three steps: BEFORE, MIDDLE and AFTER. In the first step called BEFORE (left column), subjects heard a filtered speech stimulus that lacked the key modulations for speech intelligibility. Stimuli were filtered either with a spectral modulation filter (top), removing spectral envelope modulations above 0.5 cycles/kHz or a temporal modulation filter (bottom), removing temporal envelope modulations above 3 Hz (see 'Methods' section). In the second step called MIDDLE (centre column), subjects heard the unfiltered version of the spoken sentence. A subset of three subjects had a 50% chance of hearing either the unfiltered version or pink noise with a matched frequency power spectrum. In the third step called AFTER (right column), the same filtered speech stimulus was repeated. Subjects attended to a fixation cross presented during each stimulus and passively listened to the presented sounds. (**b**) In a separate behavioural task, non-clinical subjects were asked to type any words they heard after the first filtered speech presentation (BEFORE and here labelled no context), after a filtered speech sentence that followed a different unfiltered sentence (AFTER with wrong context), or after a filtered speech sentence that followed the matching unfiltered sentence (AFTER with right context). Mean ± s.e. % words correct is shown. More details and results obtained using other contextual stimuli to further explore the stimulus information required for the perceptual enhancement can be found in Supplementary Fig. 1 and 'Methods' section.

Fig. 5B), matching the finding in control subjects that behavioural responses did not substantially change over the course of a session (Supplementary Fig. 5A).

For all Speech-R electrodes on temporal and perisylvian cortex, the mean difference in HFB activity between the BEFORE and AFTER conditions was estimated. There was a significant increase in HFB activity in the AFTER condition (cluster-based permutation test, $P = 0.003$, see Fig. 2b). This increase in activity could reflect sentence independent changes in arousal (for example, increased HFB activity to any auditory stimuli), or changes due to the activation of speech and language network resulting in a shift in gain or tuning of speech features in the degraded signal. However, only changes in tuning would lead to sentence-specific (and in our experimental paradigm, trial by trial) effects. For subjects that also had pink noise control trials, there was no difference in evoked HFB activity between the AFTER and BEFORE conditions (Supplementary Fig. 3A).

**Between-condition HFB coherence.** We next investigated whether the difference between the BEFORE and AFTER condition was only reflected in an overall increase in HFB amplitude or if there was also a difference in the time-varying details of each response. We hypothesized that HFB activity in the AFTER condition would be more similar to the activity in the MIDDLE condition (AFTER/MIDDLE) compared with the BEFORE condition (BEFORE/MIDDLE) on a trial by trial basis (that is, for individual sentences). This would provide evidence that speech-responsive electrodes responded to features in the filtered speech stimulus that were also present in the unfiltered speech context stimulus.

The time-varying coherence between the BEFORE/MIDDLE and AFTER/MIDDLE HFB activity in each trial was estimated to quantify the similarity in the responses. The between-condition

coherence for successive windows of 400 ms was calculated to evaluate the time course of evoked HFB similarity for each trial, then averaged across trials to calculate the coherence for each time bin between the BEFORE/MIDDLE and AFTER/MIDDLE conditions for each electrode over perisylvian cortex. The coherence between AFTER/MIDDLE was higher than the coherence between BEFORE/MIDDLE, indicating it was not only the mean amplitude, but also the time-varying activity that was changing from BEFORE to AFTER (permutation test, $P = 0.006$, $n = 78$; see Fig. 3, bottom row for all comparisons). These differences in coherences could still be due to an overall change: the time-varying response averaged across all trials/ sentences could be more similar to the MIDDLE (clean speech) condition in the AFTER than the BEFORE condition. This could happen if speech intelligibility resulted in simple changes in gain or if the response as measured in HFB activity to intelligible clean speech was invariant across sentences. This increase in similarity would then be reflected in individual trial responses and result in increases in our coherence estimates. However, in electrodes for which the HFB response is sensitive to spectrotemporal features of sounds, one could expect to find an additional time-varying response that is sentence-specific. To distinguish global changes from changes in sentence-specific responses, the same between-condition coherence analysis was performed after subtracting the time-varying averaged HFB response across all trials/sentences for each electrode. After subtracting this global response in each electrode, a significant increase in coherence between the responses in the AFTER/MIDDLE conditions remained (Fig. 4). This finding shows that the time-varying and sentence-specific response in each trial in the AFTER condition becomes more similar to the corresponding response to the unfiltered speech found in the MIDDLE condition and the effects are not simply due to global enhancement in neural activity.

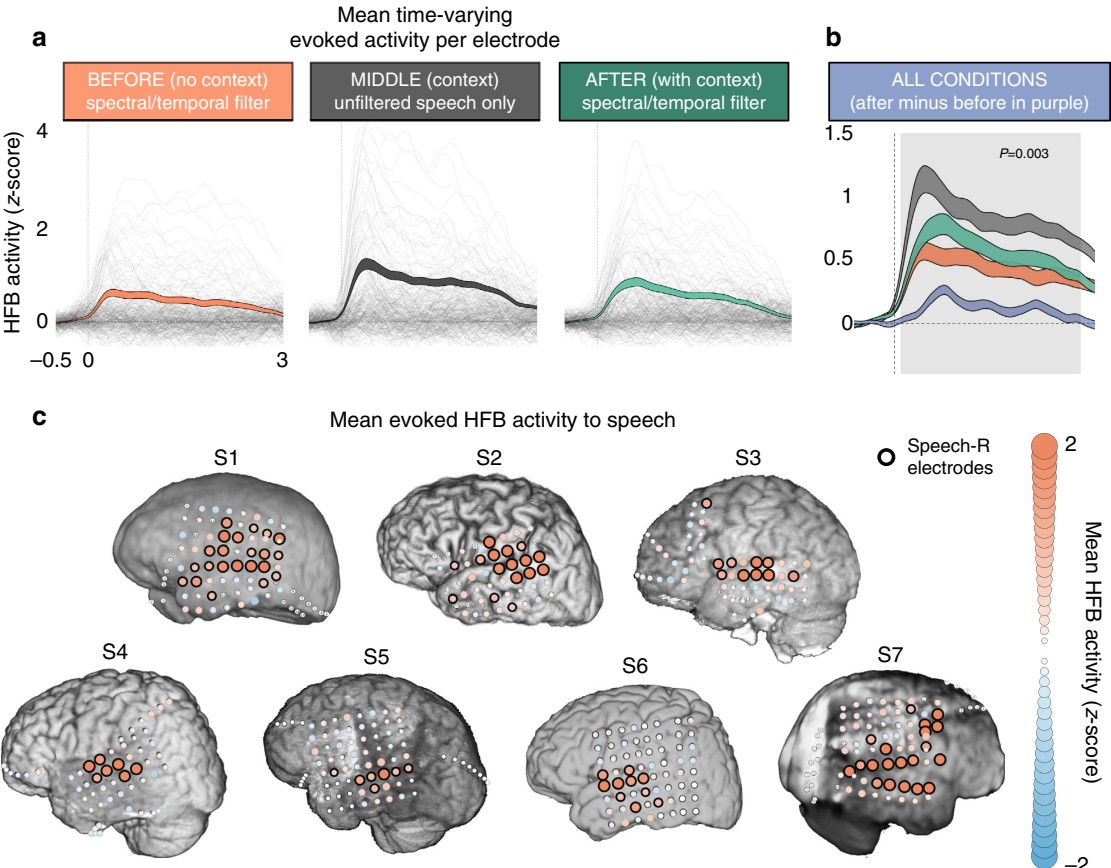

**Figure 3 | Evoked HFB activity.** (**a**) Mean HFB activity for each condition (BEFORE filtered—orange; MIDDLE unfiltered—grey; AFTER filtered—green). Individual traces are mean for each temporal lobe electrode. Shaded colour traces are grand mean ± s.e. across all electrodes. Units are $z$-scores over baseline. (**b**) Grand mean $+/-$ s.e. for all active temporal lobe electrodes in each condition. The difference between AFTER and BEFORE conditions is shown in purple. Shaded regions represent significant differences between BEFORE and AFTER conditions (cluster-based permutation test, $P = 0.003$, $n = 92$, see 'Methods' section). (**c**) Electrode coverage and average HFB activity for each subject. Electrode colours/sizes represent the mean evoked HFB activity. Dark outlines show electrodes with HFB activity significantly different from zero (two-tail permutation test, $P < 0.01$), called Speech-R electrodes. These electrodes had reliable increases in activity in response-to-speech stimuli. Speech-R electrodes located on the temporal lobe and perisylvian regions were included in eSTRF analyses.

**eSTRF modelling**. If the changes in the pattern of HFB activity in the AFTER condition relative to the BEFORE condition are related to increased speech comprehension, the AFTER activity should be more similar to the one found in response to clean speech, just as was observed. To further investigate this hypothesis, we next calculated the eSTRFs of all active temporal cortex electrodes to detect if they exhibited tuning plasticity related to an increase in speech comprehension. We estimated eSTRFs from stimulus-response (HFB) signals in each condition (BEFORE, MIDDLE and AFTER). We hypothesized that, relative to the BEFORE condition, eSTRFs in the AFTER condition would shift to be more responsive to unfiltered speech features, providing a potential mechanism for extracting speech-like features from sound and the perceptual enhancement.

eSTRF models were fit for each electrode using a jackknife approach. On each iteration one trial was left out and the model was fit on the remaining trials. The held-out trial was then used to estimate a goodness of fit (here the coefficient of determination, $R^2$) and its 99% confidence intervals. Electrodes with a confidence interval that did not overlap with 0 were considered to be electrodes 'well-modeled' by the STRF and called STRF-responsive (STRF-R) electrodes. This yielded 53 of 468 total electrodes (11.3%, see Supplementary Fig. 2). The STRF-R electrodes were also generally localized on perisylvian temporal

lobe regions (see Fig. 5 for anatomy and Fig. 6 for model score distribution).

The coefficients of each eSTRF model (that is, the spectro-temporal gains) were analysed to investigate the nature of the specific spectrotemporal tuning of each electrode. To be included in subsequent analyses, an electrode had to: (1) show evoked HFB activity in response-to-speech (Speech-R, described above and shown in Fig. 3); (2) be well-modeled by spectrotemporal features (STRF-R, described above and shown in Fig. 5), and (3) be located on the temporal lobe or perisylvian cortex, regions traditionally associated with spectrotemporal auditory processing[31,32,38]. This yielded 41 of 468 total electrodes (8.76%, see Supplementary Fig. 2).

Peaks in the eSTRFs were distributed across a wide range of frequencies, and eSTRFs were not well-characterized by simple shapes (for example, Gabor functions) as seen in typical single-unit STRFs[39,40]. This is likely due to the fact that the HFB activity represents the combined ensemble firing of many thousands of neurons in cortical columns[36] (see Fig. 7 for examples). Sparser eSTRFs have also been obtained from ECoG and single-unit data using different regularization techniques[29,41].

**Shifts in eSTRF modulation related to speech intelligibility**. To compare the spectrotemporal features present in speech with

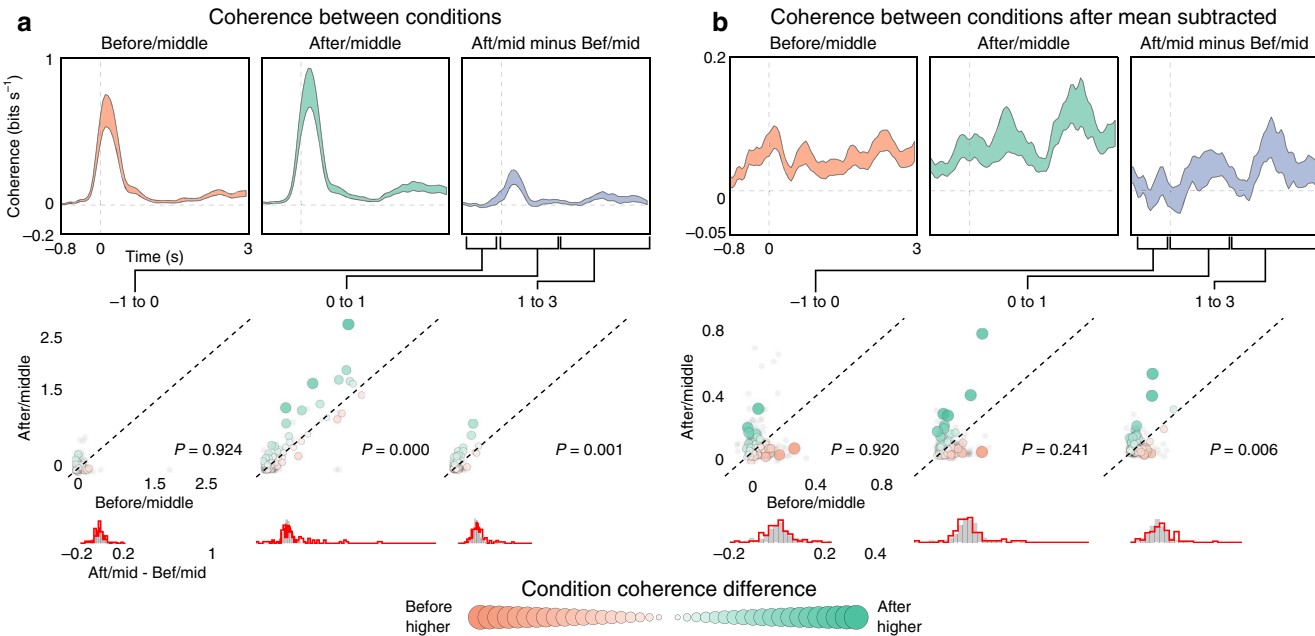

**Figure 4 | HFB similarity between conditions.** (**a**) Upper plot, time-varying integrated coherence (in bits/s, see 'Methods' section) in the evoked HFB response was calculated between pairs of conditions. Coherence was calculated for 400 ms windows in 200 ms steps. BEFORE/MIDDLE condition is shown in the left subpanels (mean ± s.e. across active electrodes), AFTER/MIDDLE condition in the middle subpanels, and the difference (AFTER/MIDDLE − BEFORE/MIDDLE) is shown on the right subpanels. Lower plot, average coherence values are shown for three time periods of interest: −1 to 0 s (left), 0 s to 1s (middle) and 1 to 2.5 s (right). Colour and size of points represent the difference AFTER − BEFORE. Inactive electrode values are shown in grey, P-value reflects the difference of condition (AFTER/MIDDLE − BEFORE/MIDDLE). Histograms below show the distribution of AFTER–BEFORE for active electrodes (red) and inactive electrodes (grey). Trial to trial coherence is higher between the AFTER/MIDDLE condition for both post-stimulus windows. (**b**). Same as in **a**, but after subtracting the average evoked response for each electrode. This compares coherence after accounting for global effects in evoked activity that are observed for all stimuli (see main text). Post-stimulus coherence is larger between the AFTER/MIDDLE conditions (permutation test for all comparisons, see bottom histograms for P-values).

those extracted by the eSTRF, we next estimated the gain of the eSTRF in the spectral and temporal modulation domain: the eSTRF modulation transfer function (MTF). The MTF shows which temporal amplitude modulations, which spectral envelope modulations, and which joint spectrotemporal modulations are emphasized (and equivalently attenuated) in the neural response. The MTF can be compared with the MPS of speech to evaluate the match in tuning between the stimulus (here speech) and the neural filters (see 'Methods' section as well as refs 40,42). The average MTF functions obtained over all of our electrodes for the BEFORE and AFTER condition are shown in Fig. 8. Qualitatively, one can observe that these MTFs are matched to the speech MPS shown in the figure. Moreover, the MTF of the shift in eSTRF (AFTER–BEFORE) averaged across all electrodes emphasizes the region of the MTF that was both preserved in the filtered speech and shown to be essential for speech intelligibility[37], suggesting that the observed eSTRF plasticity could facilitate speech perception (Fig. 8).

**Filtered speech eSTRFs increase response to speech features.** We conducted two additional analyses to quantitatively determine whether the AFTER eSTRFs became more sensitive to unfiltered speech features. First, we assessed whether the eSTRF in the AFTER condition was more sensitive to unfiltered speech features than the eSTRF in the BEFORE condition. Each filtered speech eSTRF (BEFORE and AFTER) was used to calculate a predicted response to unfiltered speech. The magnitude of this predicted response reflects the extent to which the eSTRF extracts spectrotemporal features that are present in the input stimulus, in this case unfiltered speech. The root-mean-squared power (RMS) of the output in the BEFORE and AFTER condition was

calculated and compared for each electrode: the power in the AFTER condition was higher than power in the BEFORE condition (mean RMS increase $0.12 \pm 0.03$, $P = 0.0001$, $n = 41$; see Fig. 9).

Next, we used the eSTRFs fit on the unfiltered speech MIDDLE condition to predict HFB activity in the BEFORE and AFTER conditions. The predicted HFB activity was compared with the true HFB activity to assess how well the unfiltered eSTRF characterized the mapping from acoustic features to neural activity. Larger goodness of fit ($R^2$) values indicate that the mapping of sound features onto HFB activity is more similar to that of the unfiltered speech condition. Goodness of fit scores were higher for the AFTER condition compared with the BEFORE condition (mean $R^2$ improvement $0.05 \pm 0.01$, $P = 0.0001$, $n = 41$; see Fig. 9).

Taken together these results together show that the tuning of electrodes in the AFTER condition becomes more similar to tuning acquired in response to unfiltered speech. Moreover, this shift in tuning causes the eSTRF to be more responsive to speech-like features of the stimulus.

**Filtered speech eSTRF shifts overlap with MIDDLE eSTRFs.** Finally, to directly compare the spectrotemporal tuning between conditions, we calculated the similarity between eSTRFs obtained in each condition using partial correlation (see 'Methods' section). Partial correlation measures the correlation between two variables after removing the linear relationship with a third variable. The partial correlation between the eSTRFs in the BEFORE/MIDDLE conditions was estimated after taking into account the eSTRF from the AFTER condition, and vice-versa. Partial correlations between AFTER/MIDDLE were higher than

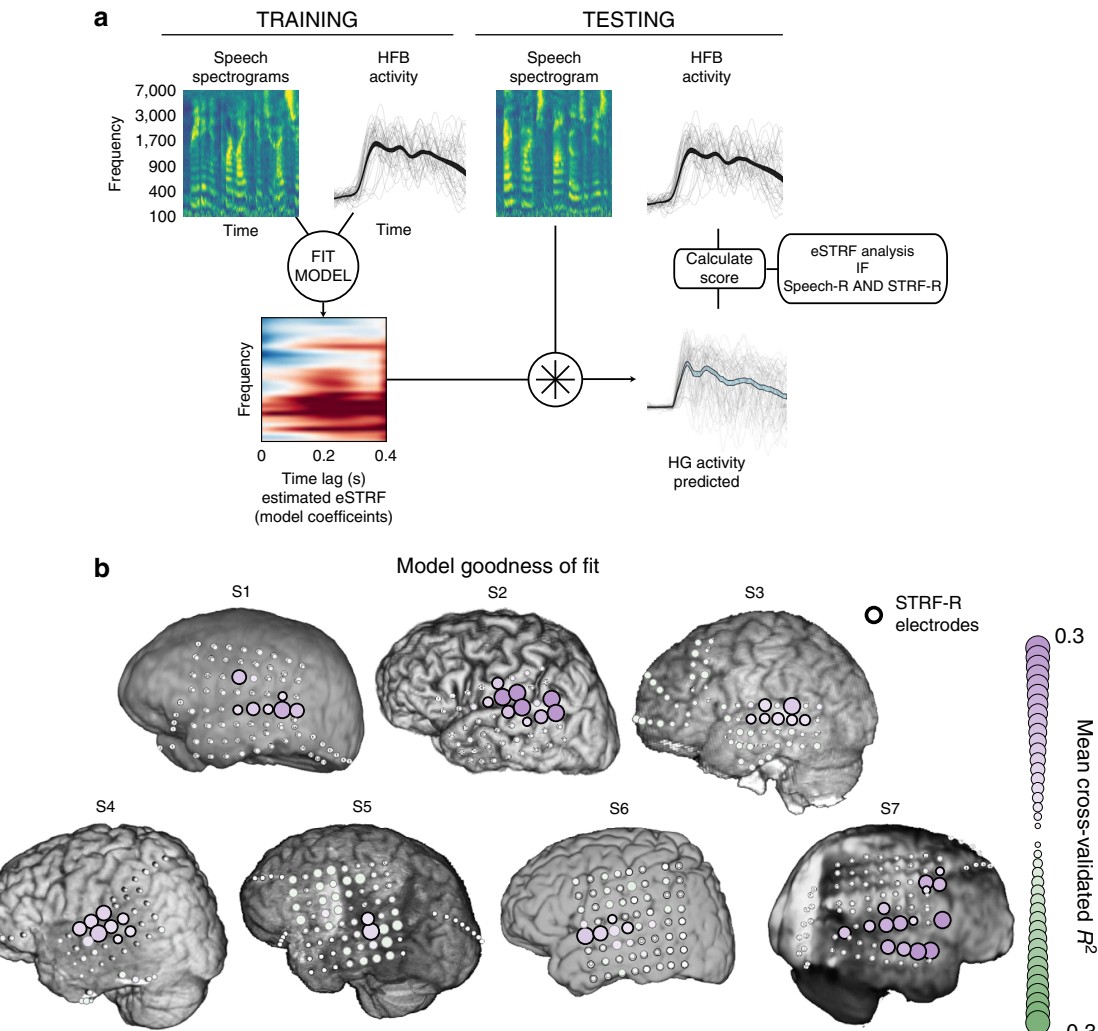

**Figure 5 | eSTRF model fitting and goodness of fit across electrodes.** (**a**) Example of model fitting procedure. Auditory spectrograms of sound and evoked HFB activity (top, first/second columns) is used to fit a linear regression model, resulting in a set of model coefficients (eSTRF, lower left). This eSTRF is convolved with a held-out auditory spectrogram (top, third column) to generated a predicted HFB activity trace (lower right). The goodness of fit (cross-validated $R^2$) is calculated between the predicted response and the actual HFB activity in the held-out trial (top, fourth column). This process is repeated, leaving out a different trial, until all trials have been included in the test set. (**b**) Average goodness of fit of the eSTRF model across subjects and electrodes. Size and colour of each electrode represents the average model score. Electrodes with black outlines had model scores significantly above 0 (confidence interval test across trials) and are designated STRF-responsive (STRF-R) and were included in further eSTRF analyses if they also showed increased HFB activity (Speech-R, see Fig. 3). See Supplementary Fig. 2 for a comparison of Speech-R and STRF-R electrodes). Negative values of cross-validated $R^2$ can occur if parts of the neural signal that aren't correlated with the stimulus spectrogram are overfit. Note that negative values of $R^2$ are small and not significantly different from zero as expected, see Fig. 6 for distribution of all $R^2$ values.

between BEFORE/MIDDLE (Fig. 10a,b; mean partial correlation improvement $0.18 \pm 0.001$; $P = 0.001$, $n = 41$, permutation test) indicating that eSTRFs obtained with degraded speech shifted to become more like the eSTRFs obtained with intelligible speech. The majority of the increase in partial correlation is located around the superior temporal gyrus (STG) (Fig. 10c), a region shown in previous research to respond to spectrotemporal features in many sounds[30–32,38]. Moreover, the increase in eSTRF similarity was itself correlated with the evoked HFB amplitude (Pearson's $r$, $P = 0.007$, see Fig. 10b). For subjects that also had pink noise control trials, there was no difference in partial correlation between the BEFORE and AFTER condition (Supplementary Fig. 3B).

**Connectivity analysis.** We also examined whether the observed eSTRF plasticity was correlated with changes in functional

connectivity measured across electrodes, providing initial clues for whether the tuning shifts could be driven by top-down effects. For this purpose, we calculated the coherence of the HFB amplitude between the electrodes included in the eSTRF analysis and groups of electrodes either in temporal or frontal/premotor cortex. There was a small significant increase in coherence in the AFTER condition relative to the BEFORE condition. Moreover, the coherence in the AFTER condition was closer to that obtained with the unfiltered speech (Supplementary Fig. 6A; Supplementary Methods). These results suggest that there may be changes in functional connectivity that are consistent with a state in the AFTER condition that is closer to the one found during the perception of intelligible speech, potentially explaining the observed changes in the tuning eSTRF for speech-like features. This amplitude coherence analysis does not, however, reveal the direction of the effect and it is also possible that the changes in

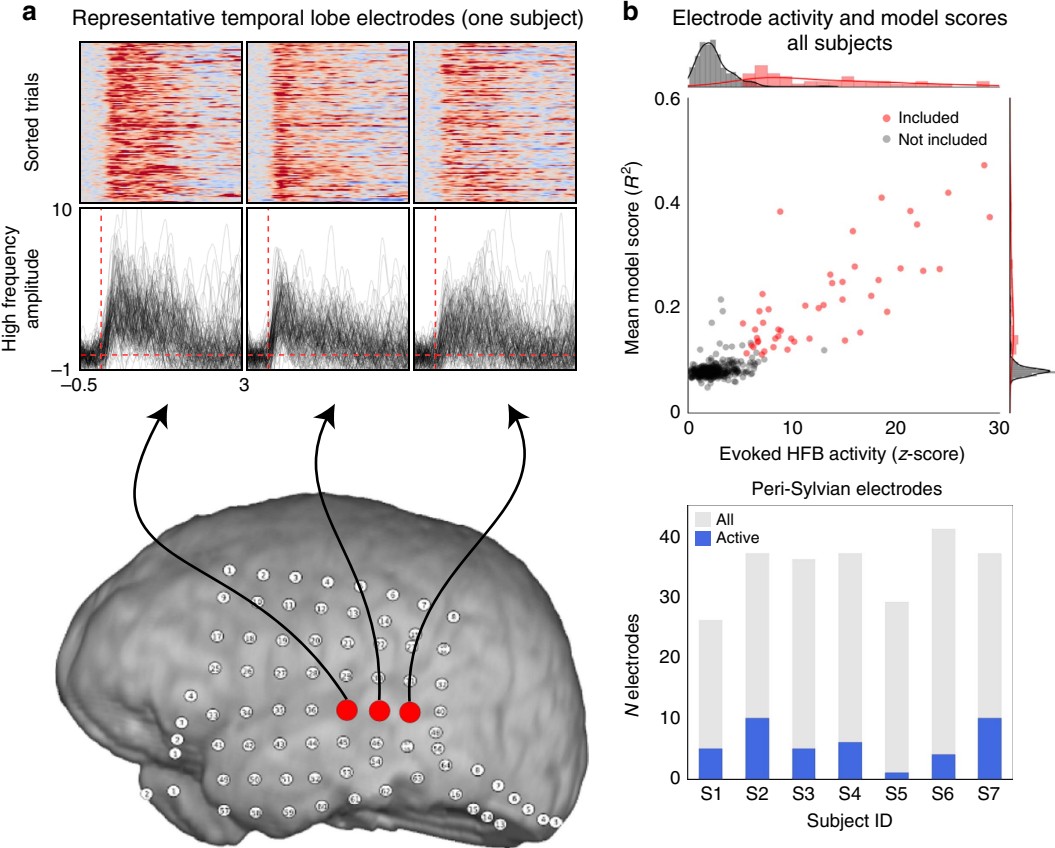

**Figure 6 | Single trial HFB activity and relation to model scores.** (**a**) Sample HFB activity for three active STG electrodes. Plots show stacked epoch plots of HFB activity, sorted by activity onset time. An increase in HFB activity is seen at the single trial level. Below, HFB activity from each trial (z-score over baseline) is shown. Temporal electrodes with high HFB activity and high model scores are plotted in red, and were included in eSTRF analyses (see 'Methods' section). Electrodes that did not meet these criteria are plotted in grey. (**b**) Top: mean HFB activity (z-score over baseline, x axis) are plotted against model scores (cross-validated coefficient of determination, $R^2$, y axis) for each electrode. Bottom: bar graph showing the number of peri-sylvian electrodes that had significant eSTRF predictions per subject.

auditory tuning cause the observed changes in functional connectivity.

To examine potential directional effects, we also calculated the phase amplitude coupling (PAC) between the phase of the ECoG signal in the 3–8 Hz from electrodes in frontal/premotor regions and the HFB amplitude for electrodes included in the eSTRF analysis in auditory cortex (Supplementary Fig. 6B). We used frequencies from 3 to 8 Hz for the phase calculation as it has been suggested that phase in this frequency range may track the envelope of a perceived speech stimulus and that the low-frequency signal could drive responses in higher frequencies[24,27,43]. In this analysis, however, we did not find significant inter-region cross-frequency activity that was modulated by task condition. Moreover, an analysis that quantified the relationship between low-frequency theta phase and the envelope of the speech utterance also did not show any differences between the BEFORE and AFTER conditions (Supplementary Fig. 7).

Thus, although changes in functional connectivity were measured both within the temporal lobe and between the frontal cortex and the temporal lobe, we are unable at this point to distinguish top-down from local or bottom-up effects. Additional experiments with greater coverage of frontal neural activity and additional analyses are required to determine the direction of the information flow that drives the observed plasticity in the temporal lobe and perisylvian region.

## Discussion

After hearing an intact sentence, subjects understand a subsequent noisy version of the same sentence that was previously unintelligible. This robust perceptual enhancement is characterized by an increase in HFB activity, onsetting within 300 ms and sustained throughout the speech utterance. Moreover, the time-varying HFB activity becomes more similar to activity during passive listening to unfiltered, intact speech, providing evidence that auditory electrodes shift how they track the time-varying properties of the filtered speech. Finally, a spectrotemporal analysis of human auditory cortical speech responses (eSTRFs) shows that the perceptual enhancement due to exposure with intact speech is paralleled by a shift in spectrotemporal tuning in auditory cortical areas. This shift in tuning overlaps with speech features, making the cortical population more responsive to unfiltered speech.

These results provide novel evidence that experience with language rapidly and automatically alters auditory representations of spectrotemporal features in the human temporal lobe. Rather than a simple increase or decrease in activity, it is the nature of that activity that changes via a shift in receptive fields. This has implications for encoding models of sound features in human/animal models, as well as in theories of top-down auditory processing. There have been attempts to characterize the neural response-to-speech under attention-based manipulations. For example, Mesgarani and Chang[32] used a decoding approach to estimate the spectral representation of sound in the

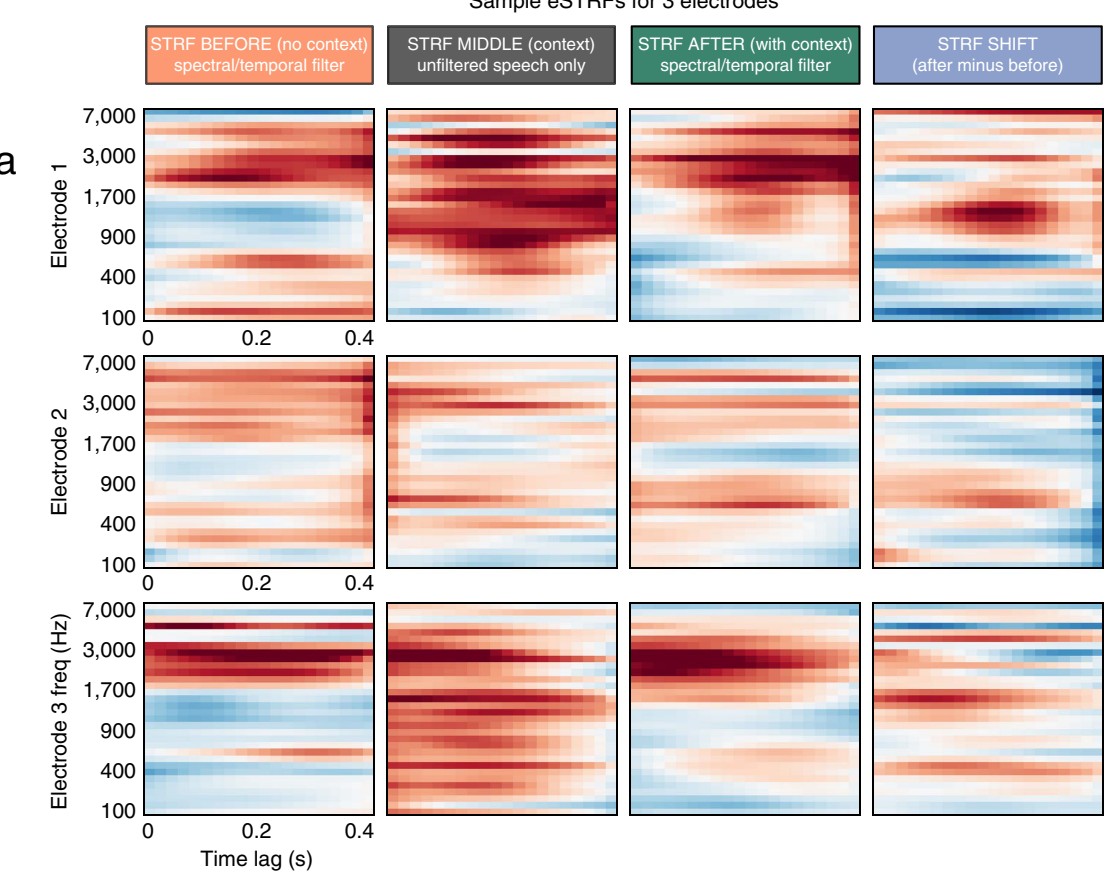

**Figure 7 | Sample eSTRFs.** (**a**) eSTRFs for multiple conditions (columns) and electrodes are shown (rows). The right column shows the change in eSTRF (AFTER–BEFORE). The gain of all eSTRFs shown has a colour scale in z-score units, where the s.d. is obtained across cross-validation folds. Subsequent analyses compare the similarity between the BEFORE/MIDDLE eSTRFs (orange) and the AFTER/MIDDLE eSTRFs (green).

auditory cortex during a task in which subjects attended to one of two utterances being played simultaneously. The authors reported that the decoded spectrogram became more similar to the speech stream that was being attended to, suggesting plasticity in the information encoded in cortical electrodes. This effect may be due to enhancing the gain of specific filter channels in the auditory cortex, as we have observed here. The tuning shift can be interpreted as a 'spectrotemporal prior' over incoming sounds, priming auditory cortical neurons to respond to particular speech-like qualities. This interpretation is compatible with higher-level theories of categorical (or probabilistic) speech representation, such as perceptual warping[44].

Relatively rapid changes in auditory STRFs have also been demonstrated in animal models. Research that showed task-dependent plasticity in auditory STRFs was initially performed in ferrets that were trained to detect target pitches in a go/no-go task[15]. More recently, it was shown that these auditory STRFs were dynamic and shifted due to concurrent top-down and bottom-up demands that depended on particular behavioural tasks[19]. This idea is supported in the current study, which revealed rapid cortical plasticity due to the knowledge of high-level auditory features. There have also been studies in animal models that report an invariance to signals embedded in different levels of background noise. For example, Rabinowitz et al.[12] showed that neurons higher in the cortical hierarchy were more invariant to noise levels. They proposed two separate adaptive gain mechanisms by which neurons separate signal from noise to be more sensitive to relevant stimulus features. Similarly, in our study, the perceptual enhancement coming from

experience with unfiltered speech could be thought of as a kind of 'signal enhancement' in which high-level information causes neurons to vary their gain to experience a signal with less noise.

How single-unit STRFs combine to form an ECoG electrode eSTRF is an important next step to bridge the gap between the animal and human literature and advance our understanding of the neural mechanisms that can drive this cortical STRF plasticity. Given the dependence of the behavioural effect on linguistic attributes, we predict that this rapid, automatic shift in the eSTRF originates at least in part from top-down signals in higher-level regions that are part of the language network such as auditory association areas[45–48], or in 'non-auditory' regions such as the inferior frontal gyrus or premotor cortices[49].

A prior ECoG study suggested that delta-theta power entraining may provide a mechanism for using temporal structure of the sound to 'chunk' relevant auditory streams and facilitate speech processing[50]. This theta power entraining might originate in prefrontal cortex and affect lower auditory areas. Although we found functional connectivity effects that were modulated by the task condition, we did not detect inter-regional PAC changes that could provide more substantial evidence for direction of the information flow. Future research with joint frontal/temporal coverage will be needed to explicate the origin of putative top-down processes (for example, from frontal regions) that might contribute to eSTRF plasticity.

In summary, in this study we demonstrate rapid spectro-temporal plasticity while subjects listened to both normal and degraded speech. We show that the human auditory cortical map is highly dynamic and context dependent, and highlight

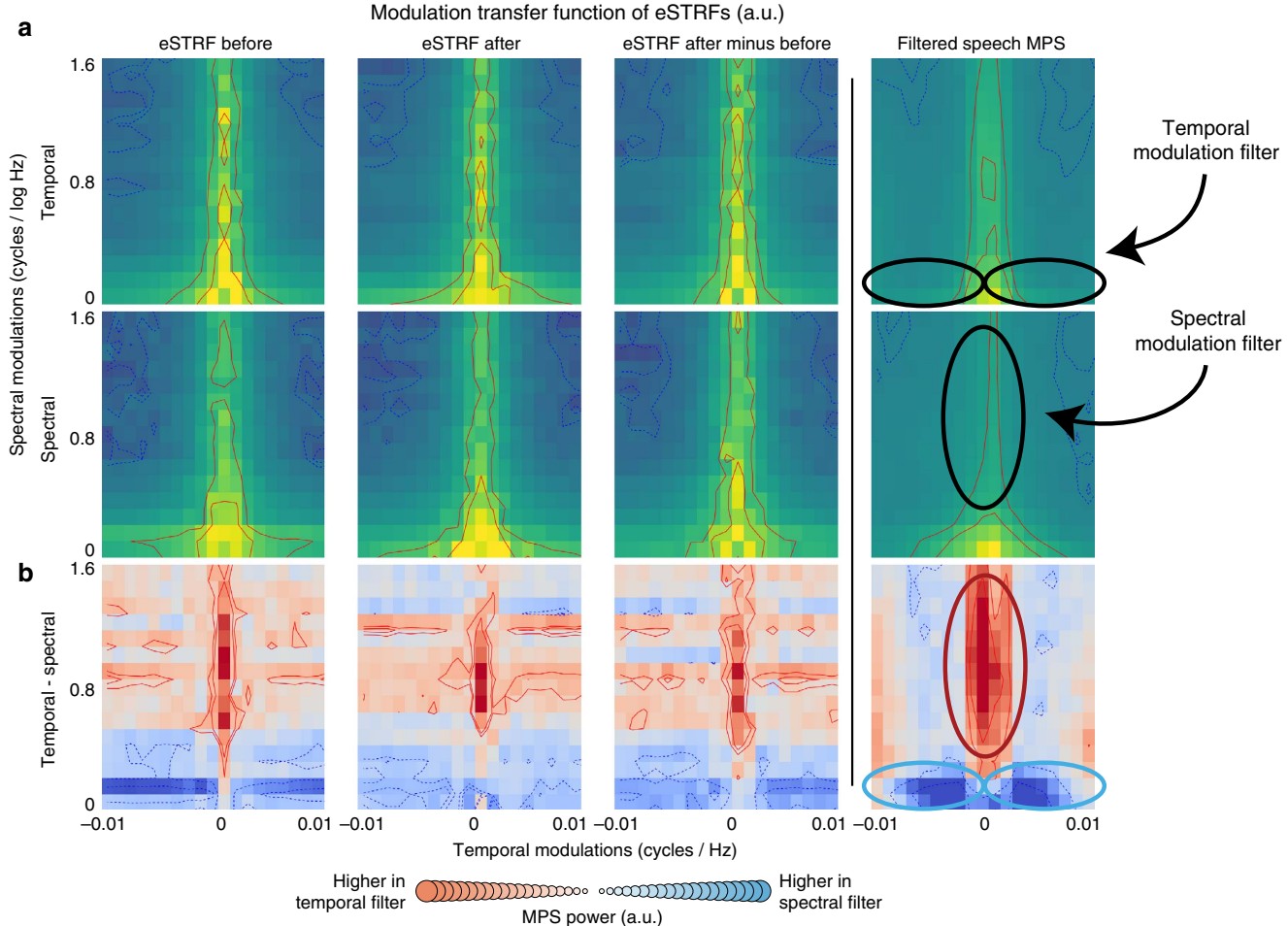

**Figure 8 | MTF of eSTRFs.** The MTF (or modulation gain) of each eSTRF was calculated, then averaged across conditions. eSTRFs were estimated separately for temporal-filtered stimuli and for spectral-filtered stimuli. (**a**) The mean MTF for eSTRFs is shown for the BEFORE (1st column) and AFTER (2nd column), condition. Lines represent 5th, 15th, 85th and 95th percentiles. The difference in eSTRF was calculated, and the MTF of this difference is shown in the 3rd column. The MPS of the actual filtered speech stimuli is shown in the 4th column for comparison. Top row are eSTRFs fit on temporal-filtered stimuli, bottom row are eSTRFs fit on spectral-filtered stimuli. (**b**) The difference in the MTF for the two filter types (temporal–spectral) was calculated for each electrode. Z-scores for this difference are shown for each condition. The MTF of each eSTRF matches its respective filter type (cluster-based permutation t-test across electrodes, $n = 41$, BEFORE $P = 0.001$, AFTER $P = 0.003$, AFTER–BEFORE $P = 0.001$), suggesting that tuning changes emphasize spectrotemporal modulations that are present in the degraded speech sound and are crucial for speech intelligibility. Note that these features were also present in the BEFORE stimulus (since BEFORE and AFTER stimuli were identical) and in the BEFORE eSTRF (as shown in the MTF for the eSTRF BEFORE, left column). This suggests that neural tuning in the BEFORE condition does match filtered speech features, and that this is accentuated after hearing the unfiltered speech.

the importance of studying sensory cortical responses with behaviourally relevant, naturalistic stimuli[2]. The dynamical changes observed in these sensory maps are dependent on a spectrotemporal prior related to high-level speech features that increases speech signal identification, enabling perception of a stimulus that was previously incomprehensible.

## Methods

**Participants and data acquisition.** Electrocorticographic (ECoG) recordings were obtained using subdural electrode arrays implanted in seven patients undergoing neurosurgical procedures for epilepsy (age 22–51; 4F/3M). Recordings took place at the University of California at Irvine (UCI), Columbia University (CU) and John Hopkins University (JH). All patients volunteered and gave their informed consent before testing, and this research was approved by the Committees for the Protection of Human Subjects at UC Berkeley, UC Irvine and the Johns Hopkins Medical School. Grid placement was determined entirely by clinical criteria (see Fig. 3 for reconstructions of subjects). Electrode grids had spacing from 5–10 mm (Adtech grids), with the following numbers of channels: JH: (48, 64), IR: (68, 68, 62), CM: (110, 104).

Multi-channel ECoG data were amplified, analog-filtered above 0.01 Hz, and digitally recorded with a sampling rate of 1 KHz (JH, CU) or 5 KHz (UCI). All

channels were subsequently down-sampled to 1 KHz, corrected for DC shifts, and band pass filtered from 0.5 to 200 Hz. Notch filters at 60, 120 and 180 Hz were used to remove electromagnetic line noise. All filters were zero-phase IIR filters implemented with the MNE-python toolbox[51]. The time series were then visually inspected to remove time intervals containing periodic spiking discharges and generalized spiking due to ictal activity. All epileptic channels, as well as channels that had excessive noise including broadband electromagnetic noise from hospital equipment and poor contact with the cortical surface, were removed from analysis. Finally, electrodes were re-referenced to a common average.

**Brain mapping of electrodes.** Each subject had post-operative anterior-posterior and lateral radiographs, as well as computer tomography (CT) scans to verify grid locations. Three-dimensional cortical models of individual subjects were generated using pre-operative structural magnetic resonance (MR) imaging. These MR images were co-registered with the post-operative CT images using Curry software (Compumedics, Charlotte, NC, USA) to identify electrode locations. Cortical activation maps were generated using custom Python software.

**ECoG filtered speech passive listening task.** ECoG subjects performed a passive listening task that consisted of 50–60 trials. In a single trial, the subject fixated on a cross in the middle of a laptop screen. Three sounds were played successively through laptop speakers. These followed the pattern 'filtered speech (BEFORE)->

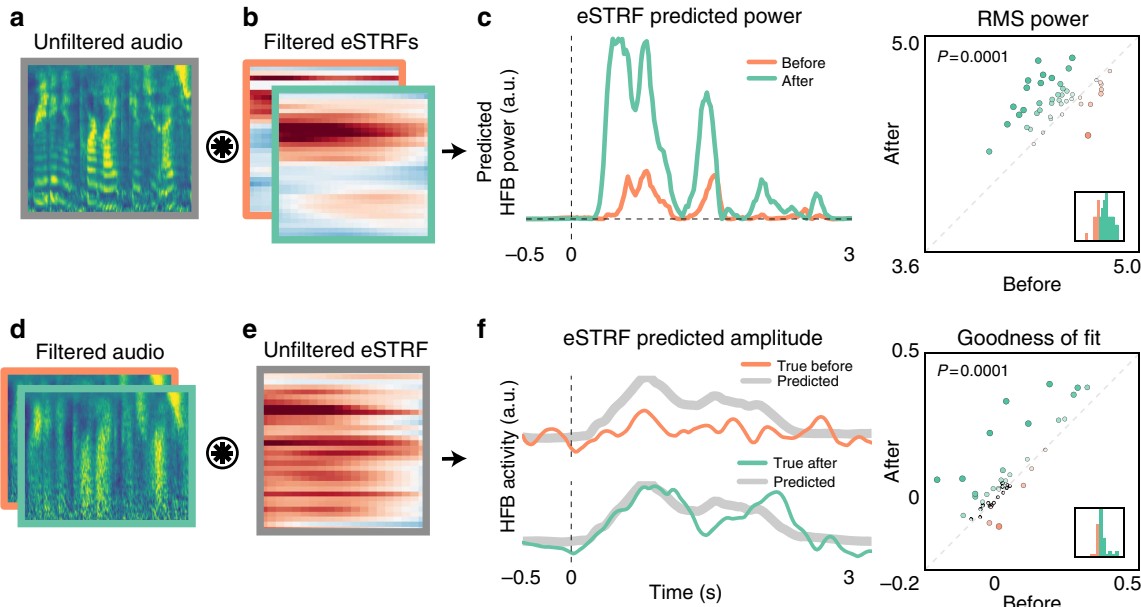

**Figure 9 | eSTRF changes overlap with speech features.** Top: spectrograms of unfiltered speech (**a**) were convolved with the eSTRF fit on each filtered speech condition (**b**). The size of the predicted response (**c**, left) depends on the overlap between the eSTRF and the unfiltered speech features. Scatterplot (**c**, right) shows the predicted response power between BEFORE and AFTER conditions. Size and colour represent the difference (AFTER−BEFORE), and inset histogram shows the distribution of differences. Bottom: spectrograms of filtered speech (**d**) were convolved with the eSTRF fit on unfiltered speech (**e**), resulting in a predicted HFB amplitude for that spectrogram (**f**, left side, grey trace). This was compared with the true HFB activity in each condition (green and orange traces). Correlation between the predicted and actual trace reflects the extent to which an unfiltered eSTRF is predictive of the neural response. The scatterplot shows the comparison between Predicted and True AFTER versus Predicted and True BEFORE (**f**, right). Size and colour of points represent the difference (AFTER−BEFORE), and inset histogram shows the distribution of this difference.

unfiltered speech (MIDDLE)-> filtered speech (AFTER)', and the speaker/content of the sentence was always the same within a single trial. However, no sentence was repeated within the same subject. Each stimulus was 2–5 s long. The inter-stimulus interval was randomly chosen between 0.5 and 1.5 s on each presentation, resulting in a trial length of 12–16 s.

In three ECoG subjects, a pink noise control trial was added to test for the effect of filtered speech repetition on electrode tuning. In these trials, the unfiltered speech context (middle sound presentation) sentence was replaced with energy-matched pink noise. This trial type made up 50% of trials in these subjects. Trials were conducted using the PsychoPy open-source toolbox[52].

**Behavioural controls for filtered speech.** Time constraints in the epilepsy ICU environment precluded detailed behavioural assessment of ECoG patients, though post-test, patients typically reported a perceptual enhancement after hearing the unfiltered speech stimuli. An additional behavioural experiment was conducted to assess the degree of perceptual enhancement after hearing the unfiltered speech using different kinds of MIDDLE context sentences. Subjects were divided into three groups. Each subject was asked to listen to speech sentences (explained below) and to type out any words they understood. The mean percentage of words for each sentence was calculated for each subject, and then compared across groups with an unpaired *t*-test. The first group ($n = 5$) replicated previous intelligibility experiments using the same stimulus set[37]. Filtered sentences were presented to subjects. After each presentation, subjects were asked to type any words that they could understand, and the per cent correct of sentence words detected was calculated.

The second group ($n = 9$) was used to control for the effect of stimulus repetition, as well as general changes in arousal due to hearing unfiltered speech. Subjects were presented with the same trial structure used in the ECoG recordings. The BEFORE/AFTER stimuli were always the same filtered speech sentence, and the MIDDLE stimuli was either an unfiltered version of the same sentence or a different sentence. Using a different sentence in the MIDDLE tests for an effect of filtered speech repetition, as no linguistic or acoustic context matches the filtered speech sentence. Using the same sentence in the MIDDLE elicits the same perceptual enhancement effect reported in ECoG subjects. Subjects were again asked to type as many words as they understood and the mean per cent correct is reported. The third group of subjects ($n = 15$) was used to test linguistic versus acoustic stimulus features on the perceptual enhancement effect. Subjects performed the same task as group 2. However, now the filtered speech context sentence was either the same sentence spoken by a different-gendered speaker, or a different sentence spoken by the same speaker. This provides a coarse split between acoustic context (same speaker, different sentence) and linguistic context (same

sentence, different speaker). Subjects were again asked to type any words they understood.

**Filtered speech sound creation.** Filtered speech was created using a MTF applied to the joint Spectral-Temporal Modulation Spectrum of the individual speech sentences as described in ref. 37. This filtering allows one to remove particular frequencies in the joint spectrotemporal envelope of the sound. Briefly, the raw sound waveform is first converted into a time-frequency representation (a spectrogram). Then, a two-dimensional (2D) Fourier transform of the sound spectrogram converts this representation into a domain that describes the spectral and temporal modulations that are present in the spectrogram.

The temporal modulations correspond to fluctuations of the amplitude envelope of the sound such as those produced by words and syllables, while the spectral modulations correspond to both coarse (such as speech formants) and fine (such as the harmonics from glottal pulse) repeated structures found along the frequency axis (see Fig. 1 for a visual explanation).

Once the sound has been transformed to this space, the gain of a large portion of frequency modulations (spectral filter) or temporal modulations (temporal filter) is set to 0 (the phase modulation spectrum is left untouched, see Fig. 1 for filtering procedure examples and Fig. 2 for spectrogram examples). This filtered MPS is converted to a spectrogram using an inverse 2D Fourier transform, and finally back into a time-varying sound wave using a recursive algorithm that selects the appropriate phase shift for each frequency band and recovers the unique sound that corresponds to that spectrogram[37]. The result is a stimulus that sounds speech-like, but is incomprehensible to the naïve listener. Note that the overall frequency power spectrum of the modulation-filtered sounds is unchanged: it is the same as the unfiltered sound.

In this study, two filters were used: a low-pass filter of spectral modulations (0.5 cycles/kHz), and a low-pass filter of temporal modulations (3 cycles/Hz). The parameters of these filters were chosen to remove respectively the spectral structure or the temporal structure that is key for speech comprehension[37].

**Neural and auditory feature extraction.** Our primary analysis consisted of fitting a linear model that predicted patterns of ECoG HFB amplitude as a function of spectral features. Auditory features (inputs to each model) consisted of time and frequency varying amplitudes based on psychoacoustic and physiological studies of language processing[3]. This 'auditory spectrogram' was obtained by estimating the amplitude envelope for 128 narrow-bands generated by a bank of erb-spaced gammatone filters ranging from 180 to 7,000 Hz. To obtain the envelope of each narrow-band signal, the output of the filter is half-wave rectified, followed by a

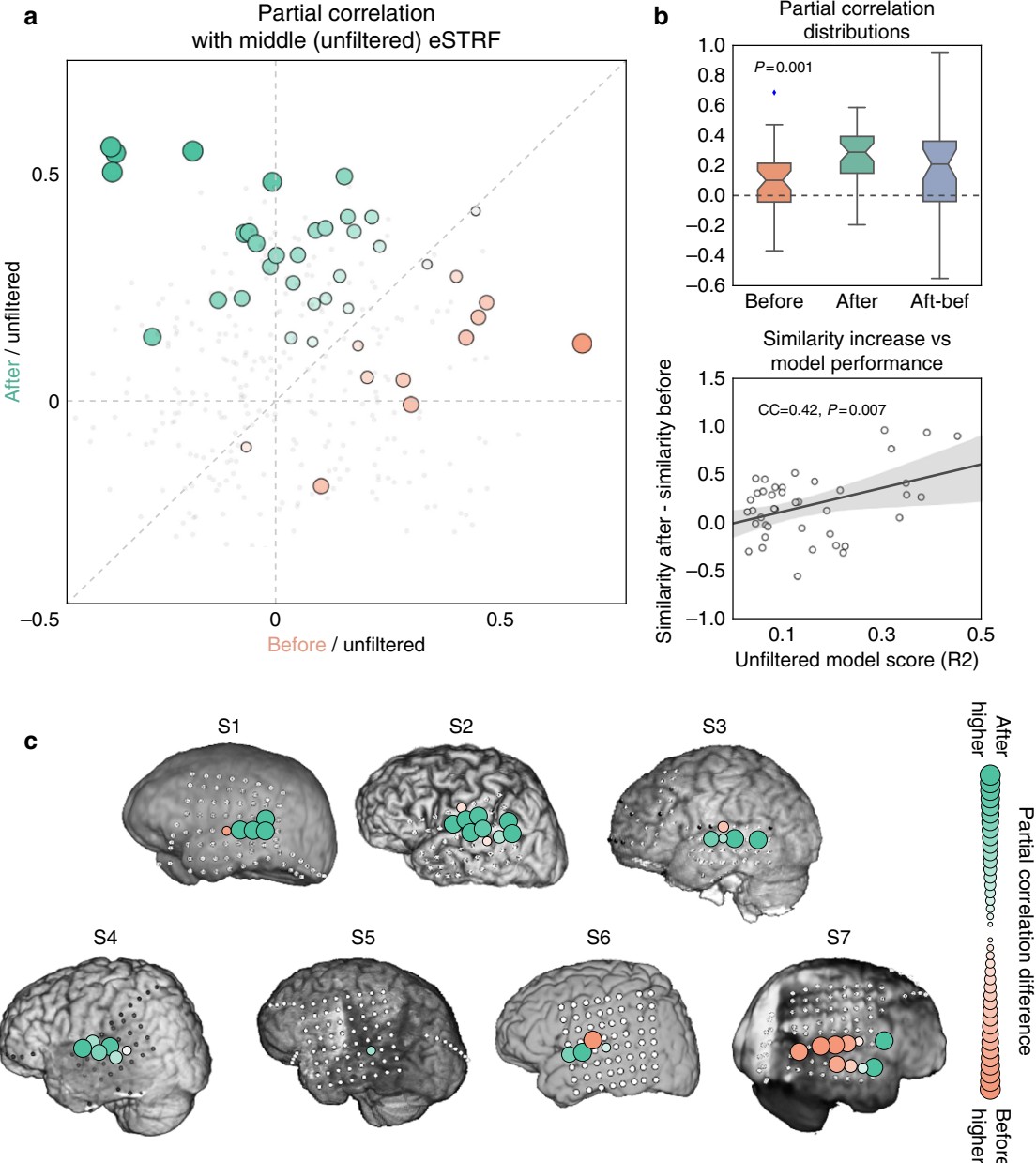

**Figure 10 | eSTRF similarity analysis. (a)** Linear similarity (partial correlation) is shown between eSTRFs for the BEFORE/MIDDLE conditions after regressing out the AFTER condition (x axis) and the AFTER/MIDDLE condition after regressing out the BEFORE condition (y axis). Electrode colour and size represent the difference in partial correlations (AFTER/MIDDLE in green, BEFORE/MIDDLE in orange). **(b)** Top: median ( ± 75th and 25th percentiles) partial correlations between MIDDLE eSTRF and the BEFORE (left bar) and AFTER (middle bar) condition, as well as difference in partial correlation between AFTER/MIDDLE and BEFORE/MIDDLE conditions (right bar). Bottom: the change in partial correlation (AFTER/MIDDLE − BEFORE/MIDDLE) increases as a function of the electrode's goodness of fit with the eSTRF model (Pearson's r, see figure for stats). **(c)** Partial correlation differences for each subject and each electrode. Model results are restricted to electrodes located on the temporal lobe. Greener colours reflect a higher partial correlation between eSTRFs in the AFTER/MIDDLE conditions. Size of the electrode represents the magnitude of the difference.

non-linear compression, and spectral sharpening. Finally, the output of each frequency band was passed through a leaky integrator with a time constant of 8 ms (details on the feature extraction can be found in ref. 3). The 128 acoustic frequencies of the initial spectrograms were subsequently down-sampled to 32 frequency bands to reduce dimensionality and computational load.

Neural activity (outputs of the model) consisted of the envelope of the HFB activity of each electrode. A window around 21 centre frequencies were defined from 70 to 140 Hz, with the width of each window increasing semi-logarithmically with frequency, following previous studies in ECoG encoding models[53]. The raw ECoG signal was first band pass filtered for each window using a zero-phase IIR filter. Then, the amplitude of the band-passed signal was calculated as the modulus of the Hilbert transform of the signal. Finally, the amplitude for each centre frequency was averaged together to attain a single time-varying estimate of HFB

activity[53]. Before estimating the linear filter, the audio spectral representation and the neural HFB response were down-sampled to 50 Hz.

**Evoked HFB and speech-responsive electrodes.** For electrode selection, we baselined each trial using times − 800 to − 100 ms relative to sound onset. We then calculated the mean post-stimulus activity in each trial. This yielded a single value for evoked HFB activity per trial/condition. For each condition, 99% confidence intervals on mean evoked activity across all trials were obtained by bootstrapping. Electrodes whose lower bound (bootstrapped 0.5th percentile) were greater than 0 in response to unfiltered speech were considered Speech-R electrodes.

To test for differences in mean HFB activity between conditions, the difference in time-varying HFB activity in each trial was calculated and then averaged across

trials to obtain a single 'difference time-varying HFB activity pattern' per electrode. Significance (the null hypothesis being no difference AFTER–BEFORE) was estimated using a cluster-based permutation test that corrects for multiple comparisons and computes statistics at the cluster level[54].

**Between-condition coherence.** The similarity in HFB activity between each filtered speech condition (BEFORE/AFTER) and the unfiltered speech condition (MIDDLE) was assessed by the measure of coherence. Coherence was chosen instead of the cross-correlation coefficient because of its robustness to high-frequency noise and invariance to systematic phase delays between signals. Similar results were obtained with correlation coefficient analysis (results not shown) but the correlation coefficient calculation requires additional assumptions on the relevant temporal scale of analysis related to the low-pass filtering needed to extract the lower frequency signals from the higher frequency noise. In the coherence calculation, the estimation of this relevant time scale is implicitly performed in a data driven manner, as the signal-to-noise is estimated for each frequency. The integral of the coherence (expressed here in bits s$^{-1}$ and shown in Fig. 4) yields then a measure of overall similarity for two time-varying signals[55]. For each stimulus, the time-varying coherence between the BEFORE/MIDDLE conditions, and between the AFTER/MIDDLE conditions was estimated using a multi-taper windowing function[56] for all the trials. The coherence was calculated for a sliding window of 400 ms moving in 200 ms steps from $-500$s to 2,500 ms, relative to stimulus onset. Unbiased estimates of the coherence for each window were obtained using a jackknife method. To compare overall coherence across electrodes, coherence was converted to normal mutual information (in bits s$^{-1}$), an information theoretic representation that allows for the integration of the coherence across frequency bands[55], which takes the following form:

$$MI_{norm} = \int_f \log_2(1 - \mathrm{coh}(f))\mathrm{d}f$$

The mean ± s.e. time-varying integrated coherence (in bits s$^{-1}$) was calculated across electrodes for each pair of conditions (BEFORE/MIDDLE and AFTER/MIDDLE). Statistics are performed for windows of interest on the mean difference in coherence between AFTER/MIDDLE and BEFORE/MIDDLE (Fig. 4). Code for performing the trial-to-trail coherence can be found in the 'Data availability' section.

**eSTRF model formulation.** Three eSTRFs were fit from the data obtained from each electrode: one using audio from the BEFORE trials, one using MIDDLE trials, and one using AFTER trials. This allowed for the comparison of eSTRFs coefficients from one trial type to the next.

The eSTRF is an encoding model that describes the linear mapping between the speech spectrogram and the HFB activity. It models the HFB signal as a weighted sum of the amplitude at each frequency band and for a range of points in time as follows:

$$\hat{R}(t, n) = \sum_{\tau} \sum_{p} g(\tau, p, n) S(t - \tau, p)$$

where $S(t - \tau, p)$ is the estimated speech representation for the frequency band $p$ at time lag $(t - \tau)$, with $\tau$ being a time lag ranging between 0 and 400 ms. $\hat{R}(t, n)$ is the estimated HFB neuronal response of electrode $n$ at time $t$. Finally, $g(\tau, p, n)$ is the linear transformation matrix (or set of eSTRFs), which depends on the time lag, feature of interest, and the electrode being predicted.

To obtain the eSTRF, a regularized linear regression algorithm was used. Linear regression attempts to find parameter values that capture the relationship between the input and output (in this case, stimulus features and brain activity). It accomplishes this by finding parameters that minimize the squared difference between model fit and training data, the minimum square error (MSE). The MSE solution is the solution that maximizes the likelihood for Gaussian noise distributions. However, when the number of model parameters is large in comparison to the fitting data size, the MSE solution can yield parameter values that are determined by the particular data set rather than the underlying relationship (called overfitting). To control for this, regression is paired with regularization, a technique that minimizes the tendency of a model to overfit data by effectively shrinking the magnitude of parameters. Shrinkage is obtained by implementing prior distributions on parameters centred at zero. This prior results in an additional penalty term that is added to the MSE.

In the case of linear ridge regression, a single parameter (here referred to as the ridge parameter) controls the penalty incurred by large parameter values. Specifically, ridge regression includes a penalty term for the L2-norm of parameter weights. This type of penalty corresponds to a Gaussian prior centred at zero for the model parameters, with the ridge parameter specifying the variance of this distribution. To choose a value of the ridge parameter, experimental trials were repeatedly split into training and test sets using a jackknife approach. On each iteration, one trial was left out for model validation. Models were fit on the training data for multiple values of the Ridge parameter. All training inputs/outputs were standardized to zero mean and unit standard deviation (that is, z-scored) before model fitting. For each model, the goodness of fit was calculated using the coefficient of determination ($R^2$) between the predicted HFB response and the actual response in the validation trial. This cross-validation was performed for all

electrodes/conditions, and repeated until all trials had been used in the test set, resulting in a distribution of selected ridge parameters yielding the maximum $R^2$.

To ensure that the prior over model coefficients was the same in all conditions, the mode of the distribution of ridge parameters for active electrodes was selected, and all models were re-fit with this single value for the ridge parameter using the same cross-validation described above. Model coefficients were averaged across all splits for final coefficient estimates. The cross-validation procedure was also used to calculate t-values of model coefficients by taking the mean divided by the standard deviation across CV splits. Code for performing encoding model fitting and cross-validation across trials can be found in the 'Data availability' section.

It should be noted that eSTRFs reported in this study visually have a slightly greater temporal extent than those in a recently published article that used a different (but related) approach to electrode receptive field analysis using maximally informative dimensions[29]. Receptive fields derived from models are sensitive to the assumptions and constraints of that model, and one would expect differences in STRF shape when using different models. This paper used L2 regularization (ridge regression) due to its interpretability, computational efficiency, and robustness and prevalence in the literature. Other alternatives such as maximally informative dimensions, boosting, or L1 (Lasso) regularization may yield sparser STRFs[29,41,57].

All model fitting was performed with custom code that relied on the Python libraries scikit-learn[58] and MNE-python[51], which are built on top of the scipy/numpy stack[59].

**MTF of eSTRFs.** To investigate whether the eSTRF gain was tuned for spectrotemporal features found in speech stimuli, the MPS of the sounds was compared with the MTF estimated for each eSTRF. Similar to a frequency power density spectrum, the MPS is obtained from the amplitude of the 2D Fourier transform of the spectrogram[42]. It shows the spectrotemporal modulations (in cycles per log-kHz for spectral modulation and in Hz for temporal modulations) that have high and low power in a given signal (corresponding to high occurrence and low occurrence). The MPS is invariant to translation and, unlike a spectrogram, can be averaged across samples of a signal to describe average properties. In a similar manner, the MTF can be obtained from the 2D Fourier Transform of a spectrotemporal filter (here the STRF) and, without averaging, shows the tuning gain of the filter in the same space as the MPS.

**STRF-responsive electrode selection.** We focused our analyses on electrodes located on the temporal lobe (particularly covering the STG and superior temporal sulcus). These regions of the brain respond to acoustic and linguistic features, and represent the best candidates for detecting a shift in spectrotemporal tuning. Responses then underwent several steps to exclude electrodes based on their non-significant and/or poorly fit responses.

The predictive score of all eSTRF models fit on unfiltered (MIDDLE) speech trials was calculated as the coefficient of determination ($R^2$ between predicted and actual HFB amplitude on held-out test data). For each electrode, we calculated the 99th percentile of model score across cross-validation splits. Electrodes whose lower bound was greater than 0 were considered spectrotemporally-responsive (STRF-R, see Fig. 6 and Supplementary Fig. 2).

To be included in the analysis, an electrode had to be Speech-R and STRF-R, and had to be located on the temporal lobe and in perisylvian regions (Supplementary Fig. 2).

**eSTRF comparisons.** To detect an eSTRF tuning shift from the BEFORE to the AFTER condition, several analyses were carried out. The goal behind each was to compare the eSTRF properties in the BEFORE and AFTER conditions to the neural response in the MIDDLE speech condition. The primary aim of all analyses is to determine whether the subjective perceptual enhancement effect corresponds to a shift in spectrotemporal tuning to filtered speech.

**MIDDLE condition coefficients generalization.** To assess the extent to which eSTRF plasticity improved the response to the speech signal, we estimated the extent to which coefficients fit in the MIDDLE condition (on unfiltered speech) generalized to the BEFORE and AFTER conditions. The eSTRF estimated in the MIDDLE condition was used to make predictions about the HFB activity in the BEFORE and AFTER condition. In this manner, one can determine the extent with which the spectrotemporal tuning estimated from unfiltered speech was a valid characterization of the tuning in each filtered condition. Predictions were obtained by convolving the eSTRF filter with the spectrogram in each filtered speech condition. We then compared the goodness of fit ($R^2$ between the predicted and actual HFB activity in the BEFORE and AFTER conditions).

**eSTRF unfiltered speech output power analysis.** Next, the extent to which eSTRFs in the BEFORE and AFTER condition are responsive to spectrotemporal features of unfiltered speech was assessed. For this purpose, we calculated the predicted HFB response to unfiltered speech using the eSTRF in the BEFORE and AFTER conditions, and as well as the output power of these predictions. This power reflects the extent to which the eSTRF overlapped with unfiltered speech

features and, thus, is able at extract unfiltered speech sounds. To account for possible changes in the overall signal-to-noise ratio of eSTRFs in all condition, each eSTRF was standardized to zero mean and unit standard deviation before convolution. Then, the unfiltered speech spectrograms were passed through the eSTRF of the BEFORE and AFTER condition. Finally, the root-mean-squared (RMS) amplitude of the output was calculated for each, and compared between the BEFORE and AFTER conditions (Fig. 9).

**eSTRF linear overlap partial correlation analysis.** For each electrode included in this analysis, the eSTRF in each condition was calculated as the $z$-score for each spectrotemporal feature across CV splits. To test the hypothesis that spectrotemporal tuning shifts after hearing the unfiltered context speech, the partial correlation was calculated between sets of conditions. Partial correlation allows one to determine the correlation between two variables, conditioned on one or more other variables. It reflects the extent to which two variables are related in a manner that is linearly orthogonal to the conditioned variables, and can be represented in the following equation:

$$\mathrm{pcorr}(a, b \mid c) = \mathrm{corr}(\widehat{a_c} - a, b)$$

where $\widehat{a_c}$ is the predicted value of variable $a$ regressed against variable $c$. In other words, one calculates partial correlation by regressing out all conditional variables, and then calculating the correlation between the residuals and the variable of interest. Partial correlation was calculated between the BEFORE/MIDDLE models and the AFTER/MIDDLE models with the following convention:

$$\mathrm{sim}_{(\mathrm{bef,\ mid})} = \mathrm{pcorr}(\mathrm{before,\ middle} \mid \mathrm{after})$$
$$\mathrm{sim}_{(\mathrm{aft,\ mid})} = \mathrm{pcorr}(\mathrm{after,\ middle} \mid \mathrm{before})$$

One would expect to find many similarities between the brain activity in response to the BEFORE and AFTER condition. As such, one wants to control for these similarities when calculating the correlation between BEFORE/AFTER and the unfiltered condition. Partial correlation allows one to determine the extent to which one condition is correlated with the unfiltered condition, after removing the linear relationship with the other condition.

**Between-condition permutation test statistics.** The following permutation-based procedure was used to compare a statistic for the difference between conditions. For each electrode, the statistic of choice (for example, HFB amplitude or partial correlation) was computed for each condition. Then, the difference between conditions for each electrode was calculated. A null condition effect of condition would have values distributed around 0. Because of the paired nature of this design, one may simulate a permuted null distribution by randomly flipping the sign of all difference values, effectively randomizing values to condition A or B. The mean of the permuted difference vector was calculated as a single point in the null distribution. This procedure was repeated 10,000 times to construct a null distribution against which the 'true' difference vector mean is compared. Reported $P$-values are the quantile for the difference vector mean with respect to this null distribution. All statistical tests are two-sided.

**Data availability.** Raw data is stored in the Collaborative Research in Computational Neuroscience (CRCNS) database at UC Berkeley (crcns.org). It can be accessed with a free CRCNS account at crcns.org/data-sets. This manuscript relied heavily on the Python packages MNE-python, scikit-learn, numpy, scipy, pandas and matplotlib. Analyses were conducted using these packages, and the large majority have been aggregated as a python package hosted on github. Code for performing statistical permutation tests is found in the MNE-python statistics module. Code for model fitting, feature extraction, statistics, and visualization can be found at github.com/choldgraf/ecogtools.

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

## Acknowledgements

We would like to thank Fionnuala Howell and Rebecca Krasnoff for running the behavioural studies, Matar Haller for brainstorming analysis and troubleshooting ideas, Elena Golumbic for assisting with data collection, and Benedicte Rossi for providing anatomical brain sketches. This study was supported by the NDSEG Graduate Fellowship, NINDS 2R37NS021135, NIDCD R01 007293, DFG SFB-TRR 31 and the Nielsen Corporation.

## Author contributions

C.R.H. created behavioural task and plan for analysis, collected data, wrote code for analysis, analysed data, and is principal author of manuscript. W.d.H. created behavioural task. B.P. created behavioural task, assisted with analysis. J.R. assisted with analysis. J.J.L. and N.C. managed patients and surgery. R.T.K. created behavioural task, helped write the manuscript, assisted in interpretation. F.E.T. created behavioural task, assisted with analysis, helped write the manuscript and assisted in interpretation.

## Additional information

**Competing financial interests:** The authors declare no competing financial interests.

