## [Peer Review File · Nature Communications]

Editorial Note: this manuscript has been previously reviewed at another journal that is not operating a transparent peer review scheme. This document only contains reviewer comments and rebuttal letters for versions considered at *Nature Communications*. Mentions of prior referee reports have been redacted.

Reviewers' comments:

Reviewer #1 (Remarks to the Author):

This is a resubmission of a manuscript following a review for [redacted]. The authors recorded high-frequency ECoG activity in auditory cortex of passively listening patients during presentation of distorted and undistorted speech. After subjects had heard the undistorted speech, neural responses to the distorted speech changed to enhance the speech signal. These changes parallel improved perception following exposure to the undistorted speech

The authors have done a good job addressing the concerns raised in the previous review. In particular, they have demonstrated that exposure to undistorted speech produces plasticity in cortical representations such that subsequent responses to noisy speech more closely resemble responses to the undistorted speech than before that exposure.

This is a complex manuscript that addresses challenging open questions about speech processing by humans. The results are novel and interesting. However, the writing could be made more clear. More specifically, the authors should provide the reader with a better interpretation of the results:

1. How does/might the observed plasticity support enhanced speech representations? The coherence analysis in Fig. 3 shows enhanced signal-to-noise levels, similar to studies that used less complex distortions (Rabinowitz et al 2013, Mesgarani et al 2014). This signal enhancement is presumably the result of changes in neural filter properties, but what is the strategy? Should a simple shift in the eSTRF toward the undistorted condition produce this effect (Fig. 6)? This logic doesn't quite follow, since simply shifting toward the undistorted condition will not selectively amplify signals over noise. Supplementary figs 4-5 may give some clues, but this analysis is buried quite deep, and the results are not really explained in a broader context. A Discussion that makes some attempt to synthesize will make the ms much more relevant to a broad audience.

2. Minor point: For the analysis in Fig. 3, what happens if a simple cross-correlation is used rather than coherence? eSTRFs will be the same only inasmuch as response phase is preserved across trials, while coherence allows phase to differ. It seems like correlation is a better match to the eSTRF, and so it would be interesting to know if the after-middle similarity also holds up for this more traditional metric.

Lesser concerns:

Line 25: "join" should be "joint"

Line 123: 67.7% : Fig 1B suggested nearly 100% words reported correct for the same condition. Are these numbers for different pools of subjects?

Line 145: Supplemental Figure 11 is referenced but there is no Sup. Fig. 11. Please check numbering everywhere as there may be some other mismatches.

Line 222: A relevant citation here might be Hulleit et al J Neurosci 2016. The eSTRFs reported for STG in this study appear to be more narrowband, presumably reflecting different analytical techniques. Some comment/comparison would be helpful.

Lines 243-261: The results are presented in the opposite order in the text than in the panels in Fig. 7. This is confusing. The results are also presented quite telegraphically and an extra couple sentences explaining why they support the conclusion of better noisy speech coding would help.

Line 550: The equations for partial correlation are rather abstract. Can the numerical equation be provided to help support replicability?

Editor's note: Reviewer 1 also commented on Reviewer 2 and 3's comments:

Comments on Reviewer 2's comments:

Reviewer 2's concerns mostly overlapped with mine (Reviewer 1's), and they appear to be addressed adequately.

Just some minor comments on Supp Fig 3, relevant to R2 comment #1:

Panel B, top: The dot sizes in the legend don't match the plot. Coupled with luminance changes from opacity of overlapping dots, it's hard to interpret at first glance.

Panel B, bottom: To address R2' question, would be nice to also see the fraction of temporal lobe channels that met criterion for inclusion rather than just the number N.

Reviewer 1's comments on Reviewer 3's comments:

The replies to reviewer #3 seem adequate. The authors added several new and relevant analyses of top-down signals, which produced some intriguing suggestions, such as the increased HFB inter-electrode coherence during the After condition. A deeper exploration top-down signals would be interesting, but is out of the scope of the current data set.

The results themselves are clearly novel with respect to existing studies on plasticity of auditory coding in humans, and the authors make that clear.

Just a small typo (missing word?) at the end of Supp. Fig. 7 legend:
"mean PAC for the 4-8Hz in each condition"

Overview

Reviewer

This is a resubmission of a manuscript following a review for [redacted]. The authors recorded high-frequency ECoG activity in auditory cortex of passively listening patients during presentation of distorted and undistorted speech. After subjects had heard the undistorted speech, neural responses to the distorted speech changed to enhance the speech signal. These changes parallel improved perception following exposure to the undistorted speech

The authors have done a good job addressing the concerns raised in the previous review. In particular, they have demonstrated that exposure to undistorted speech produces plasticity in cortical representations such that subsequent responses to noisy speech more closely resemble responses to the undistorted speech than before that exposure.

This is a complex manuscript that addresses challenging open questions about speech processing by humans. The results are novel and interesting. However, the writing could be made more clear. More specifically, the authors should provide the reader with a better interpretation of the results:

RESPONSE

We thank the reviewer for his/her constructive criticism. We have followed most of his/her recommendations and have reorganized the results section to improve the logic of the analyses performed and their interpretation. We believe that the reviewer will find a clearer paper and that all of his/her reservations have been addressed. We have organized this document such that reviewer comments are in italics, followed immediately by our response to those comments. In addition, we have highlighted relevant sections in the manuscript / supplemental material.

Major Points

Reviewer

1. How does/might the observed plasticity support enhanced speech representations? The coherence analysis in Fig. 3 shows enhanced signal-to-noise levels, similar to studies that used less complex distortions (Rabinowitz et al 2013, Mesgarani et al 2014).

RESPONSE

As mentioned above, we have reorganized the paper to more explicitly show how our results and analyses support plastic receptive fields for enhanced plasticity. The response to this first point will become clearer after reading our detailed responses below to all the sub questions and points raised by the reviewer, starting here with the use of coherence.

As the reviewer notes, the coherence analysis shows an enhanced “clean-speech” signal in the after condition. We concur that we did not adequately explain the meaning of this observation. The increase in coherence can be thought of as an increase in SNR, but note that this is not in the traditional sense of the signal becoming more reproducible across trials (relative to noise), as is commonly reported in the STRF literature (for example, to assess the goodness of fit of STRFs as we have done in the past¹). We believe that the reviewer understood this, but we have altered the text to be more explicit and clear. (see section *Between-condition HFB coherence*, as well as *Methods: Between-condition coherence*)

The coherence analysis performed here shows that the signal in the *After* conditions (distorted speech) becomes more similar to the signal in the clean speech *Middle* condition. This is true when averaging across all sentences as well as on a sentence to sentence basis. The first plot in the figure shows the “average” coherence analysis. This increase in similarity could be due to an overall increase in gain in the ECoG signal when speech is understood and/or when arousal levels are higher. To clarify these possibilities, we conducted the second analysis shown in the figure, which subtracts the time-varying average before calculating coherence. We also found an increase in coherence on a sentence by sentence basis, above and beyond the effect observed with the mean response. Thus, in the *After* condition, the specific time-varying profile of the ECoG signal in response to each distorted sentence becomes more similar to the ECoG activity obtained in response to unfiltered speech.

Although this does not prove that this response plasticity supports speech enhancement, it is what one would expect in such a situation. Specifically, if speech perception to some extent depends on the processing measured in auditory cortical areas, then neural responses (here *After/Middle* relative to *Before/Middle*) would be expected to be more similar if the auditory representation is more similar. We have updated the text to clarify the points made above, and to more directly link it with subsequent analyses of STRF properties that show that indeed the tuning has changed to extract speech features. In those later analyses we also cite the work by Rabinowitz et al 2013 as well as Mesgarani and Chang 2014^{2,3}.

Reviewer

This signal enhancement is presumably the result of changes in neural filter properties, but what is the strategy? Should a simple shift in the eSTRF toward the undistorted condition produce this effect (Fig. 6)? This logic doesn't quite follow, since simply shifting toward the undistorted condition will not selectively amplify signals over noise. Supplementary figs 4-5 may give some clues, but this analysis is

buried quite deep, and the results are not really explained in a broader context. A Discussion that makes some attempt to synthesize will make the ms much more relevant to a broad audience.

RESPONSE

The reviewer makes a good point that changes in the eSTRF towards the clean speech condition does not necessarily mean that these changes lead to speech enhancement. For example, the eSTRF (in the clean speech condition) could be tuned to acoustic features that are not particularly important for speech comprehension, and a shift in that direction would not necessarily lead to increased discriminability (see our previous comments on the coherence analysis as well).

To investigate the extent to which eSTRF changes enhanced the extraction of speech signals we improved explanation/discussion of the two additional STRF analyses also reported in the original paper but not properly described or organized. The first one (in the main text) quantifies the amplitude of the output when passing clean speech through the eSTRF calculated in both the *Before* and the *After* condition. The amplitude is significantly larger for *After* eSTRFs, showing that the shift from *Before* to *After* leads to bigger predicted responses to clean speech. This analysis shows that the eSTRF in the *After* condition has more gain for speech features than the eSTRF in the *Before* condition. It is discussed more clearly in the section *Filtered speech eSTRFs become more responsive to speech features*, and is shown in Figure 7, top row.

The second analysis (originally in the supplemental material as Figures S4/S5, now in the main text as Figure 6 and Supplementary Figure S4) examines the tuning shift in the modulation space of the eSTRFs. This analysis shows that the tuning shift is towards spectral-temporal features that remained in the distorted speech signal, and that are known to be important for speech⁴. This second analysis is more convincing visually and more explicitly shows how the eSTRF tuning shifted, but we thought is potentially less accessible to a non-specialist. For these reasons and for space limitations, it was originally only mentioned in the supplemental material. We now realize that it is an important part of our arguments and that it belongs in the main text *before* the quantitative measure obtained by examining the power in the prediction to clean speech. This is now found in the Results Section: *eSTRF gain shifts for spectro-temporal modulations are important for speech intelligibility*, and in Figure 6.

Thus, we agree with the reviewer that the above analyses and results should be presented together in the main paper. In order to improve the logic behind our conclusions, we have re-ordered the results presented in the paper and attempted to connect them more logically according to the reviewer's suggestions.

We first discuss how eSTRFs are fit, visualize them, and discuss their tuning properties (e.g., their Modulation Transfer Function or MTF). The visualization of the shifts in the tuning MTF and the visual comparison with the speech Modulation Power Spectrum, clearly and intuitively shows how this plasticity allows increases in neural gain for features of the sound that are present in speech and important for speech comprehension. This is found in the section *eSTRF gain shifts for spectro-temporal modulations are important for speech intelligibility*. Next, to quantify how much gain for speech features is achieved, we present the analysis of eSTRF output amplitude in response to unfiltered speech. That analysis shows an

RMS increase, reflecting increased eSTRF tuning to speech features. This is now found in the Results Section: *Filtered speech eSTRFs become more responsive to speech features*. Finally, we present the eSTRF partial correlation analysis to link the increase in output amplitude to an increased in overlap in eSTRF features between the *After* and *Middle* conditions. This is now found in the Results Section: *Spectrotemporal tuning shifts overlap with unfiltered speech eSTRFs*. We believe that this follows the logic of the paper's argument more clearly.

In addition to the results section, we have also modified our discussion section in order to discuss the "meaning" behind the eSTRF shift, and to connect it more strongly with the single unit literature. There are few receptive field plasticity studies in humans, but there is an extensive literature in animal models which we believe is relevant to this work. We've discussed this in more detail, citing literature in STRF plasticity as well as noise invariance as suggested by the reviewer, and have tried to make more clear avenues for future research. We believe this is in the spirit of the reviewer's suggestion, since a goal of this manuscript is to make connections between human and animal literature on sensory tuning.

Reviewer

2. Minor point: For the analysis in Fig. 3, what happens if a simple cross-correlation is used rather than coherence? eSTRFs will be the same only inasmuch as response phase is preserved across trials, while coherence allows phase to differ. It seems like correlation is a better match to the eSTRF, and so it would be interesting to know if the after-middle similarity also holds up for this more traditional metric.

RESPONSE:

We believe that coherence is a more appropriate measure for examining correlations in time series. First, as mentioned by the reviewer, it is not sensitive to particular phase delays (i.e. time shifts) as long as these are reproducible across trial (the coherence as we calculated does average the cross-correlations across trials and thus requires fixed delays). This is beneficial in a situation where the two signals being compared do not have a single, stable phase delay. When using the cross-correlation coefficient, these phase delays are taken to be zero, which may not be the case if the gain changes in the eSTRFs are the result of different sub-populations of neurons responding to the incoming (and occasionally delayed) top-down signals of varying temporal properties. One could calculate a full cross-correlation function that shows the average cross-products for different delays and use the delay that gives the peak in the cross-correlation. That value could be normalized by the square root of the variance of each signal but this is not a traditional measure in time series analysis. The coherence (or its time equivalent the coherency) yields the correct normalization.

The other advantage of coherence (obtained with correct normalization in the Fourier domain) over correlation is that it evaluates the similarity between the signals at all time scales, i.e. identical to estimating correlations in time with varying time-windows or low-pass filters of the

data. The alternative would be to filter the signal (here low-pass filter) to focus on frequencies with maximum correlations, and remove the frequencies that are just noise. For these two reasons, we elected to use coherence in order to capture the general similarity between the two signals, regardless of their relative phase or the time window. An overall “correlation” can then be obtained by effectively integrating the coherence function over all frequencies (i.e. relevant time windows). We can define the similarity between two time series as a single information theoretic quantity, which we have calculated in the paper and now explicitly define in the methods (see sections on *Between-condition coherence* in main paper and methods).

This said, the results obtained with a cross-correlation coefficient with a smoothing of 5Hz and at zero delays show the same effect (now discussed in the methods section *Between-condition coherence*). Below, we’ve included the results using cross-correlation with a fixed lag of 0 (this was the maximum lag at which all condition comparisons had maximum correlation). They are similar to those presented in the manuscript, though we would rather report the more powerful measure of coherence.

Condition comparison, correlation coefficient

Lesser concerns

Reviewer

Line 222: A relevant citation here might be Hullett et al J Neurosci 2016. The eSTRFs reported for STG in this study appear to be more narrowband, presumably reflecting different analytical techniques. Some comment/comparison would be helpful.

RESPONSE

This is clearly a very relevant recent paper that was not published at the time of our first submission. Briefly, the differences in STRF structure could be due to several factors. Firstly, in Hullett et al, the authors use a different method for STRF calculation based on Maximally Informative Dimensions. This method may have different properties for sparsity of STRF features, which would affect how smooth / peaked the STRFs look. The authors performed similar analyses with Ridge regression (what we have used here), but do not show any figures, data, or code with the Ridge regression STRF results, which makes it difficult to compare directly. We have included a discussion of this paper, as well as a more general comment on how model choice affects STRF properties in the results and methods in the main text. (see *eSTRF modeling* in main text and *Methods: eSTRF model formulation*)

Small Items

Line 25: "join" should be "joint"

DONE

Line 123: 67.7% : Fig 1B suggested nearly 100% words reported correct for the same condition. Are these numbers for different pools of subjects?

We have revised the text and figure to reflect the correct numbers reported. The figure scaling was slightly off, and we have updated it to more clearly reflect the effect sizes reported.

Line 145: Supplemental Figure 11 is referenced but there is no Sup. Fig. 11. Please check numbering everywhere as there may be some other mismatches.

We have now linked references to figures to the legends listed at the bottom of the manuscript in order to keep their labeling/numbering consistent.

Lines 243-261: The results are presented in the opposite order in the text than in the panels in Fig. 7. This is confusing. The results are also presented quite telegraphically and an extra couple sentences explaining why they support the conclusion of better noisy speech coding would help.

We've re-ordered these two points in the results section. We've also added more explanatory material to transition into this section (and re-ordered to make it more clear).

Line 550: The equations for partial correlation are rather abstract. Can the numerical equation be provided to help support replicability?

We've added an equation to explain partial correlation in terms of regression out other variables in the methods section.

 Editor's note: Reviewer 1 also commented on Reviewer 2 and 3's comments:

Just some minor comments on Supp Fig 3, relevant to R2 comment #1:

Panel B, top: The dot sizes in the legend don't match the plot. Coupled with luminance changes from opacity of overlapping dots, it's hard to interpret at first glance.

We have updated this figure to keep the legend size the same as the scatterplot, and to ensure consistency between the two scatterplot colors shown.

Panel B, bottom: To address R2' question, would be nice to also see the fraction of temporal lobe channels that met criterion for inclusion rather than just the number N .

We have updated this figure to include both the total number of peri-sylvian electrodes, as well as the number of active peri-sylvian electrodes for each subject.

Reviewer 1's comments on Reviewer 3's comments:

Just a small typo (missing word?) at the end of Supp. Fig. 7 legend:
"mean PAC for the 4-8Hz in each condition"

DONE

Reviewer Response References

1. Hsu, A., Borst, A. & Theunissen, F. E. Quantifying variability in neural responses and its application for the validation of model predictions. *Netw. Comput. Neural Syst.* **15**, 91–109 (2004).
2. Rabinowitz, N. C., Willmore, B. D. B., King, A. J. & Schnupp, J. W. H. Constructing Noise-Invariant Representations of Sound in the Auditory Pathway. *PLoS Biol.* **11**, (2013).
3. Mesgarani, N. & Chang, E. F. Selective cortical representation of attended speaker in multi-talker speech perception. *Nature* **485**, 233–6 (2012).
4. Elliott, T. M. & Theunissen, F. E. The modulation transfer function for speech intelligibility. *PLoS Comput. Biol.* **5**, e1000302 (2009).

REVIEWERS' COMMENTS:

Reviewer #1 (Remarks to the Author):

The revisions have clarified important points about the nature of changes in tuning before and after exposure to the clean speech condition. The changes substantially strengthen the authors' conclusions and make their results much easier to understand. The manuscript now provides novel and compelling insight into the mechanisms underlying human perception of speech in noise.